# LEARNING TO REASON WITHOUT EXTERNAL REWARDS

**Xuandong Zhao**[*]
UC Berkeley
xuandongzhao@berkeley.edu

**Zhewei Kang**[*]
UC Berkeley
waynekang07@gmail.com

**Aosong Feng**
Yale University
aosong.feng@yale.edu

**Sergey Levine**
UC Berkeley
svlevine@berkeley.edu

**Dawn Song**
UC Berkeley
dawnsong@berkeley.edu

## ABSTRACT

Training large language models (LLMs) for complex reasoning via Reinforcement Learning with Verifiable Rewards (RLVR) is effective but limited by reliance on costly, domain-specific supervision. We explore Reinforcement Learning from Internal Feedback (RLIF), a framework that enables LLMs to learn from intrinsic signals without external rewards or labeled data. We propose INTUITOR, an RLIF method that uses a model's own confidence—termed *self-certainty*—as its sole reward signal. INTUITOR replaces external rewards in Group Relative Policy Optimization (GRPO) with self-certainty scores, enabling fully unsupervised learning. Experiments demonstrate that INTUITOR matches GRPO's performance on mathematical benchmarks while achieving better generalization to out-of-domain tasks like code generation, without requiring gold solutions or test cases. Our findings show that intrinsic model signals can drive effective learning across domains, offering a scalable alternative to RLVR for autonomous AI systems where verifiable rewards are unavailable. Code is available at https://github.com/sunblaze-ucb/Intuitor.

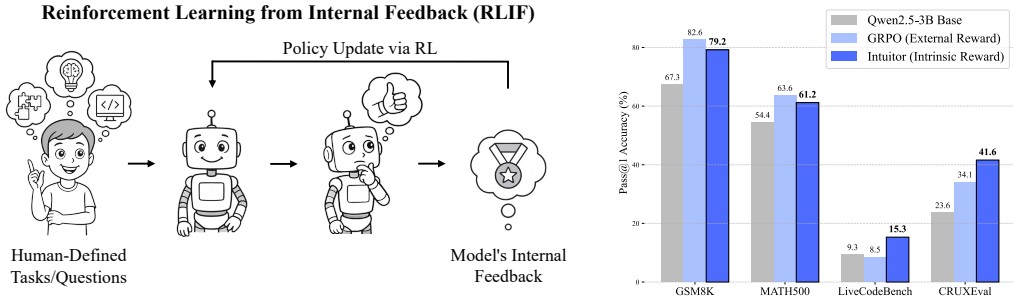

Figure 1: Overview of RLIF and INTUITOR's Performance. Left: RLIF, a paradigm where LLMs learn from intrinsic signals generated by the model itself, without external supervision. Right: Performance comparison of Qwen2.5-3B Base, GRPO, and INTUITOR (our RLIF instantiation). Both GRPO and INTUITOR are trained on the MATH dataset. INTUITOR achieves comparable performance to GRPO on in-domain mathematical benchmarks (GSM8K, MATH500) and demonstrates better generalization to out-of-domain code generation tasks (LiveCodeBench-v6, CRUXEval). Part of the illustration was generated by GPT-4o.

## 1 INTRODUCTION

Reinforcement learning has become essential for enhancing large language model capabilities. Early work focused on Reinforcement Learning from Human Feedback (RLHF), which aligns model outputs with human values through reward models trained on preference data (Ouyang et al., 2022).

---

[*]Equal contribution.

Recent advances in Reinforcement Learning with Verifiable Rewards (RLVR) replace learned reward models with automatically verifiable signals, such as exact answer matching in mathematical problem-solving, demonstrating improved reasoning capabilities in models like DeepSeek-R1 (Guo et al., 2025; Lambert et al., 2024).

Despite these successes, both RLHF and RLVR face fundamental limitations that constrain their broader applicability. RLHF requires extensive human annotation, making it expensive and potentially biased (Gao et al., 2023). RLVR, while avoiding learned reward models, demands domain-specific verifiers and gold-standard solutions. In mathematics, this requires expert annotation of solutions; in code generation, it necessitates comprehensive test suites and execution environments (Liu et al., 2023; Liu & Zhang, 2025; Team et al., 2025; Xiaomi, 2025). These requirements limit RLVR to carefully curated domains and complicate deployment in open-ended scenarios. Moreover, outcome-oriented verifiable rewards limit transferability to other domains. These challenges motivate exploration of more general and scalable reward paradigms, leading to a critical research question: *Can LLMs enhance their reasoning abilities by relying solely on intrinsic, self-generated signals, without recourse to external verifiers or domain-specific ground truth?*

In this paper, we introduce and explore such a paradigm: *Reinforcement Learning from Internal Feedback (RLIF)*, where models optimize intrinsic feedback to improve performance without external rewards or supervision. The motivation for RLIF extends to future scenarios where models develop superhuman capabilities that become difficult for humans to evaluate directly (Burns et al., 2023), requiring self-improvement through intrinsic mechanisms (Oudeyer & Kaplan, 2007).

Under the RLIF paradigm, we propose INTUITOR, a novel reinforcement learning approach leveraging a model's own confidence as an intrinsic reward. This builds on observations that LLMs exhibit lower confidence on difficult problems (Farquhar et al., 2024; Kuhn et al., 2023; Kang et al., 2024; 2025); optimizing for confidence should improve reasoning capabilities. Specifically, we use self-certainty (Kang et al., 2025), the average KL divergence between the model's output distribution and a uniform distribution, as our confidence measure. This metric has proven useful for distinguishing high-quality responses from flawed ones (Kang et al., 2025; Ma et al., 2025). Building on this insight, INTUITOR guides learning through self-generated signals, eliminating the need for external supervision or handcrafted rewards. The implementation of INTUITOR is simple, efficient, and effective: we replace the verifiable reward signal in existing RLVR frameworks, specifically Group Relative Policy Optimization (GRPO) (Shao et al., 2024), with self-certainty scores, using the same policy gradient algorithm.

Our experiments demonstrate promising results. On the MATH dataset (Hendrycks et al., 2021) with Qwen2.5-3B base (Yang et al., 2024), INTUITOR matches the performance of GRPO without relying on any gold answers. As INTUITOR rewards the generation trajectory rather than only the end result, it generalizes more effectively: training a Qwen2.5-3B base model on MATH yields a 65% relative improvement on LiveCodeBench Code generation task (Jain et al., 2024) versus no improvement for GRPO, and a 76% gain on CRUXEval-O (Gu et al., 2024) compared with 44% for GRPO. Additionally, when we fine-tune the Qwen2.5-1.5B base model with INTUITOR on the MATH corpus, a model that originally produces repetitive content and scores 0% on LiveCodeBench learns to emit coherent reasoning chains and well-structured code, reaching 9.9% accuracy after the tuning. Beyond the Qwen family, experiments with Llama (Meta AI, 2024) and OLMo (OLMo et al., 2024) models also show impressive gains, underscoring the strong generalization capabilities of INTUITOR. As INTUITOR requires only a clear prompt and no verifiable reward, it is broadly applicable across tasks, providing fresh evidence that pretrained LLMs possess richer latent behavioral priors than previously recognized.

Our contributions can be summarized as follows:

- We introduce and explore Reinforcement Learning from Internal Feedback (RLIF), a novel reinforcement learning paradigm enabling LLMs to improve reasoning skills by leveraging intrinsic, self-generated signals, without reliance on external supervision or labeled data.
- We introduce INTUITOR, an RLIF-based method that utilizes a model's own internal confidence measure—termed *self-certainty*—as the sole intrinsic reward.
- We demonstrate that INTUITOR matches supervised RL performance on in-domain tasks and achieves competitive, sometimes better out-of-domain generalization. We uncover emergent structured reasoning and enhanced instruction-following capabilities induced by intrinsic rewards.

## 2 RELATED WORK

**Reinforcement Learning from Human Feedback (RLHF).** RL has become instrumental in refining LLMs. Early pivotal work centered on Reinforcement Learning from Human Feedback (RLHF) (Ouyang et al., 2022), which aligns LLMs with human values by training a reward model on human preference data. While effective, RLHF is often resource-intensive due to the need for extensive human annotation (Touvron et al., 2023). Subsequent innovations like Direct Preference Optimization (DPO) (Rafailov et al., 2023) aimed to simplify this by directly training models on preferences. The reliance on human-generated or model-approximated human preferences poses scalability challenges and introduces potential biases from the reward model itself (Gao et al., 2023).

**Reinforcement Learning with Verifiable Rewards (RLVR).** RLVR emerged as a powerful alternative, particularly for tasks with clear correctness criteria like mathematical reasoning and code generation (Hu et al., 2025; Team et al., 2025; Xiaomi, 2025). RLVR utilizes rule-based verification functions, such as exact answer matching (Guo et al., 2025; Team et al., 2025; Xiaomi, 2025; Jaech et al., 2024), to provide reward signals, thereby avoiding the complexities and potential pitfalls of learned reward models. This approach has sparked significant advances, with models like DeepSeek-R1 (Guo et al., 2025) achieving state-of-the-art reasoning capabilities. The development of robust policy optimization algorithms like GRPO (Shao et al., 2024) and its variants (Luo et al., 2025; Liu et al., 2025) has further solidified RLVR's success. Nevertheless, RLVR's applicability is largely confined to domains where verifiable gold solutions or exhaustive test cases can be constructed, and its predominant focus on outcome-based rewards can limit generalization to dissimilar tasks or those requiring nuanced, process-oriented feedback.

**Intrinsic Signals and Self-Play in LLM Optimization.** Self-play and intrinsic rewards enable autonomous model improvement. Methods like SPIN (Chen et al., 2024) and Self-Rewarding LMs (Yuan et al., 2024) use the model itself for feedback. Earlier work like STaR (Zelikman et al., 2022) relies on outcome evaluation, while others explore procedural generalization (Poesia et al., 2024; Cheng et al., 2024). Concurrent works such as Genius, TTRL, SRT, and Absolute Zero (Xu et al., 2025; Zuo et al., 2025; Shafayat et al., 2025; Zhao et al., 2025) leverage unlabeled queries for RL but are often restricted to specific task distributions. Song et al. (2025) examine LLM self-improvement through the generation–verification gap, while Huang et al. (2025) study it through the lens of sharpening dynamics. INTUITOR aligns with this direction, offering a lightweight, general-purpose approach using self-certainty as a confidence-based intrinsic reward, enabling single-agent RL across diverse tasks without explicit feedback or gold labels.

## 3 METHOD

### 3.1 REINFORCEMENT LEARNING FROM INTERNAL FEEDBACK (RLIF)

To overcome the limitations of RLHF's costly human annotation and RLVR's domain-specific supervision, we propose Reinforcement Learning from Internal Feedback (RLIF). Instead of depending on external evaluation, RLIF uses the model's own assessment of its outputs as feedback. This offers several advantages: it reduces reliance on supervision infrastructure, provides task-agnostic reward signals, and supports learning in domains where external verification is unavailable. The optimization objective for policy $\pi_\theta$ is:

$$\max_{\pi_\theta} \mathbb{E}_{o \sim \pi_\theta(q)} \left[ u(q, o) - \beta \mathrm{KL}[\pi_\theta(o|q) \| \pi_{\mathrm{ref}}(o|q)] \right] \tag{1}$$

where $q$ is an input query, $o$ is the generated output, $\pi_{\mathrm{ref}}$ is an initial reference policy, and $\beta$ controls the KL divergence to prevent excessive deviation from $\pi_{\mathrm{ref}}$. Here, $u(q, o)$ is an intrinsic signal derived from the model's internal state or computation, rather than external verification. The key challenge lies in identifying intrinsic signals that correlate with output quality and can effectively guide learning.

Concurrent research explores related concepts within the RLIF paradigm. For example, Entropy Minimized Policy Optimization (EMPO) (Zhang et al., 2025) minimizes LLM predictive entropy on unlabeled questions in a latent semantic space. SEED-GRPO (Chen et al., 2025) uses the semantic entropy of generated sequences, combined with ground truth rewards, to modulate policy updates.

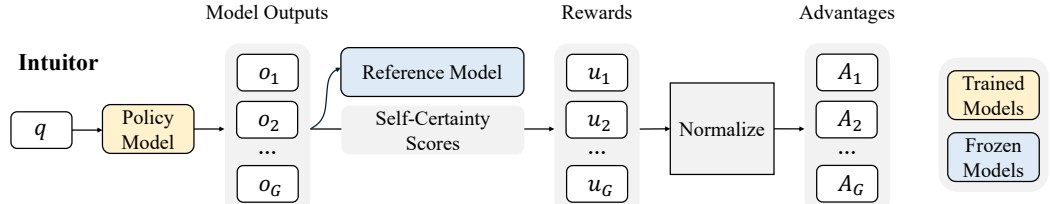

Figure 2: INTUITOR simplifies the training strategy by leveraging self-certainty (the model's own confidence) as an intrinsic reward to incentivize reasoning abilities without external supervision.

Reinforcement Learning with a Negative Entropy Reward (EM-RL) (Agarwal et al., 2025) employs a reward signal based solely on the negative sum of token-level entropy, akin to REINFORCE but without labels. These methods highlight the growing interest and potential of leveraging intrinsic signals for LLM training under the RLIF framework.

## 3.2 INTUITOR: POLICY OPTIMIZATION WITH SELF-CERTAINTY

We propose INTUITOR, a novel RLIF method that utilizes a model's own confidence as the sole intrinsic reward signal $u(q, o)$. Our choice of model confidence as the intrinsic reward is motivated by observations that LLMs often exhibit lower confidence when encountering unfamiliar tasks or lacking sufficient knowledge (Kang et al., 2024). Conversely, higher confidence frequently correlates with correctness. By rewarding increased self-confidence, INTUITOR encourages to iteratively "practice" and refine its reasoning pathways until it becomes more confident in its outputs.

We adopt the self-certainty metric from Kang et al. (2025), defined as the average KL divergence between a uniform distribution $U$ over the vocabulary $\mathcal{V}$ and the model's next-token distribution:

$$\textbf{Self-certainty}(o|q) := \frac{1}{|o|} \sum_{i=1}^{|o|} \text{KL}(U \parallel p_{\pi_\theta}(\cdot|q, o_{<i})) = -\frac{1}{|o| \cdot |\mathcal{V}|} \sum_{i=1}^{|o|} \sum_{j=1}^{|\mathcal{V}|} \log\left(|\mathcal{V}| \cdot p_{\pi_\theta}(j|q, o_{<i})\right)$$

where $o_{<i}$ are the previously generated tokens and $p(j|q, o_{<i})$ is the model's predicted probability for token $j$ at step $i$. Higher self-certainty values indicate greater confidence.

Self-certainty is related to a KL divergence where the model's prediction is the second argument, $\text{KL}(U \parallel p_{\pi_\theta})$. This contrasts with entropy (or reverse KL divergence from uniform). Critically, self-certainty is reported to be less prone to biases towards longer generations, a common issue with perplexity or entropy-based measures (Fang et al., 2024; Kang et al., 2025), making it a potentially more reliable indicator of intrinsic confidence. Kang et al. (2025) demonstrate that self-certainty is effective for selecting high-quality answers from multiple candidates, and uniquely among different confidence measures, its utility improves with more candidates. Optimizing for self-certainty thus encourages the model to generate responses that it deems more convincing. The RL process can achieve this by, for instance, guiding the model to produce more detailed reasoning steps, thereby increasing the model's conviction in its final answer. This mechanism is more nuanced than simply increasing the probability of the most likely output; it involves modifying the generation process itself to build confidence.

To optimize the objective in Equation 1, various policy gradient algorithms can be employed. Informed by the recent success in models such as DeepSeek-R1 (Guo et al., 2025) and its widespread adoption of GRPO in open-source projects, we utilize GRPO to optimize for self-certainty. The overall pipeline for this GRPO-based instantiation of INTUITOR is illustrated in Figure 2.

The core idea behind the optimization is to sample multiple candidate outputs for a given query and use their relative rewards to estimate advantages for policy updates. For each query $q \sim P(Q)$, GRPO samples a group of $G$ outputs $o_1, \ldots, o_G$ using a behavior policy $\pi_{\theta_{old}}$ (e.g., a previous iteration or the SFT model). The target policy $\pi_\theta$ is then optimized by maximizing:

$$\mathcal{J}_{\text{GRPO}}(\theta) = \mathbb{E}_{\substack{q \sim P(Q), \\ \{o_i\}_{i=1}^{G} \sim \pi_{\theta_{old}}(\cdot|q)}} \left[ \frac{1}{G} \sum_{i=1}^{G} \frac{1}{|o_i|} \sum_{t=1}^{|o_i|} \left( \min\left[ c_{i,t}(\theta)\hat{A}_{i,t}, \ \text{clip}_\varepsilon\left(c_{i,t}(\theta)\right)\hat{A}_{i,t} \right] - \beta\mathbb{D}_{\text{KL}}\left(\pi_\theta \| \pi_{\text{ref}}\right) \right) \right]$$

where $c_{i,t}(\theta) = \frac{\pi_\theta(o_{i,t}|q,o_{i,<t})}{\pi_{\theta_{\text{old}}}(o_{i,t}|q,o_{i,<t})}$ is the importance weight, $\text{clip}_\varepsilon$ is the function that clips to $[1 - \varepsilon, 1 + \varepsilon]$. Hyperparameters $\epsilon$ (for clipping) and $\beta$ (for KL penalty strength) control stability and exploration, and $\hat{A}_{i,t}$ is the advantage estimate.

**Integration of Self-Certainty.** The key innovation in INTUITOR is replacing external rewards with self-certainty scores in GRPO's advantage computation. Specifically, each output $o_i$ is scored by:

$$u_i = \text{Self-certainty}(o_i|q), \quad \hat{A}_{i,t} = \frac{u_i - \text{mean}(\{u_1, u_2, \cdots, u_G\})}{\text{std}(\{u_1, u_2, \cdots, u_G\})}. \quad (2)$$

This formulation enables the policy to favor outputs that the model itself considers more confident. The complete INTUITOR training pipeline operates by sampling multiple candidate outputs for each query, computing self-certainty scores for each candidate, using these scores to estimate advantages within the group, and updating the policy to increase the likelihood of generating high-confidence outputs. This process requires no external supervision, making it a self-reinforcing learning loop.

## 4 EXPERIMENTAL SETUP

**Training Setup.** Both GRPO and INTUITOR are trained with the Open-R1 framework (Face, 2025) on the training split of the MATH dataset (Hendrycks et al., 2021), which contains 7,500 problems. We use Qwen2.5-1.5B and Qwen2.5-3B (Yang et al., 2024) as backbone models, with a chat-based prompting format throughout. Given the models' initially weak instruction-following abilities, we do not require them to disentangle intermediate reasoning from final answers. Each update processes 128 problems, generating 7 candidate solutions per problem, with a default KL penalty of $\beta = 0.005$. For a fair comparison, GRPO and INTUITOR share identical hyperparameters (see Appendix) without additional tuning. We also evaluate a GRPO variant, denoted GRPO-PV in Table 1, which uses plurality voting[1] as a proxy for ground truth. This follows the approach from TTRL (Zuo et al., 2025), which shows that self-consistency-based rewards can match the performance of golden answers when training on inference data.

**INTUITOR for Code Generation (INTUITOR-Code).** To assess generalization beyond mathematical reasoning, we apply INTUITOR to the Codeforces code generation dataset (Li et al., 2022). This variant, denoted INTUITOR-Code in Table 1, modifies the setup as follows: the number of sampled completions per problem is increased to 14; the learning rate is reduced from $3 \times 10^{-5}$ to $1 \times 10^{-5}$; and the KL penalty is increased to $\beta = 0.01$. For simplicity, we limit the run to 50 steps, utilizing a total of 3,200 problems.

**Evaluation.** Evaluations generally use the same chat-style prompting format as in training, except for MMLU-Pro (Wang et al., 2024), where we follow the benchmark's original prompt format. Greedy decoding is used for all completions. Experiments were conducted on NVIDIA A100 GPUs, each with 40GB of memory. We evaluate performance on the following benchmarks (1) *Math reasoning*: MATH500 and GSM8K, using the `lighteval` library (Habib et al., 2023). (2) *Code reasoning*: CRUXEval-O (Gu et al., 2024), using the `ZeroEval` framework (Lin, 2024), and LiveCodeBench v6 (LCB) (Jain et al., 2024). (3) *Instruction following*: AlpacaEval 2.0 with length-controlled win rates (Dubois et al., 2024), judged by GPT-4.1 (OpenAI, 2025).

## 5 RESULTS AND ANALYSIS

We evaluate the effectiveness of INTUITOR by addressing the following research questions:

- (RQ1) How does the overall performance of INTUITOR compare to supervised RLVR methods?
- (RQ2) How does intrinsic feedback influence the model's qualitative behavior?
- (RQ3) How robust is online self-certainty when used as an intrinsic reward signal during training?

Table 1 presents main evaluation results, and Figure 3 illustrates response length evolution during training. On in-domain MATH and GSM8K datasets, INTUITOR and GRPO-PV (both golden-answer-free) achieve performance comparable to GRPO (using golden answers). This aligns with

---

[1]Self-consistency uses a plurality rule, selecting the most frequent answer even without majority support, while majority voting requires $> 50\%$ support and otherwise yields no winner (De Condorcet et al., 2014).

Table 1: Performance comparison of various methods on reasoning and instruction-following benchmarks. INTUITOR-Code is trained on Codeforces data with a smaller learning rate and fewer training steps. All evaluations are obtained with the chat inference template, except for MMLU-Pro.

| Model | Training Data | GSM8K | MATH500 | LCB | CRUX | MMLU-Pro | AlpacaEval |
|---|---|---|---|---|---|---|---|
| *Qwen2.5-1.5B Results* | | | | | | | |
| Base | - | 0.002 | 0.090 | 0.000 | 0.000 | 0.297 | 2.10 |
| + GRPO | MATH | 0.747 | 0.560 | 0.056 | 0.328 | 0.315 | 4.03 |
| + INTUITOR | MATH | 0.711 | 0.530 | 0.099 | 0.296 | 0.310 | 4.28 |
| *Qwen2.5-3B Results* | | | | | | | |
| Base | - | 0.673 | 0.544 | 0.093 | 0.236 | 0.377 | 3.72 |
| + GRPO | MATH | 0.826 | 0.636 | 0.085 | 0.341 | 0.403 | 6.91 |
| + GRPO-PV | MATH | 0.820 | 0.636 | 0.086 | 0.299 | 0.398 | 6.17 |
| + INTUITOR | MATH | 0.792 | 0.612 | 0.153 | 0.416 | 0.379 | 7.10 |
| + INTUITOR-Code | Codeforces | 0.743 | 0.572 | 0.153 | 0.411 | 0.386 | 4.16 |

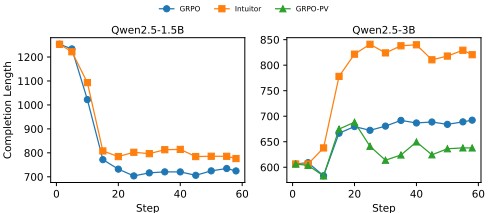

Figure 3: **INTUITOR encourages longer, more detailed reasoning during training.** This figure shows the average response length during training rollouts on the MATH dataset. For Qwen2.5-1.5B, INTUITOR and GRPO reduce gibberish outputs. For Qwen2.5-3B, INTUITOR and GRPO increase reasoning length; INTUITOR yields significantly longer responses. GRPO-PV shows minimal length increase.

Figure 4: **Mastery of in-domain skills facilitates subsequent generalization to new domains.** This figure plots the performance evolution on MATH500 (in-domain, left) and Live-CodeBench (out-of-domain, right) for models trained on the MATH dataset. MATH500 accuracy increases rapidly at first, preceding gains in code-generation accuracy. LiveCodeBench performance continues to rise even after MATH500 accuracy plateaus.

TTRL (Zuo et al., 2025), where plurality voting approximated golden answers without significant performance loss. While INTUITOR performs slightly worse than GRPO overall, on MATH it produces longer responses and demonstrates markedly improved code generation, suggesting enhanced reasoning capabilities.

## 5.1 LEARNING TO FOLLOW INSTRUCTIONS

INTUITOR significantly enhances instruction-following. Initially, the pretrained Qwen2.5-1.5B struggles with chat-style prompts, scoring <10% on all chat-template tasks (Table 1) and generating repetitive, nonsensical output, which inflates average response lengths (Figure 3). Fine-tuning with INTUITOR sharply reduces such gibberish, decreases completion lengths, and enables non-trivial performance across all evaluated benchmarks. Furthermore, on the MATH dataset, INTUITOR substantially improves the Length Control Win Rate on AlpacaEval for both Qwen2.5-1.5B and Qwen2.5-3B, surpassing GRPO under identical settings. This demonstrates robust gains in instruction adherence.

Table 2: **INTUITOR demonstrates faster initial learning compared to GRPO.** This table shows the in-domain performance on GSM8K and MATH after only 10 training steps. In all cases, INTUITOR achieves higher accuracy than the GRPO baseline, which uses ground-truth rewards.

| Model | Method | GSM8K | MATH |
|---|---|---|---|
| Qwen2.5-1.5B | Baseline | 0.002 | 0.090 |
| | GRPO | 0.081 | 0.296 |
| | INTUITOR | **0.152** | **0.368** |
| Qwen2.5-3B | Baseline | 0.673 | 0.544 |
| | GRPO | 0.758 | 0.596 |
| | INTUITOR | **0.811** | **0.618** |

## 5.2 FOSTERING STRUCTURED REASONING

**Rapid Initial Learning.** Self-certainty, a continuous and inherently process-aware reward derived from the model's internal assessment across all tokens, contrasts with binary rewards. This internal signal may encourage LLMs to follow more effective learning trajectories. Given comparable final performance between GRPO and INTUITOR, we assess early-stage learnability by comparing in-domain accuracy at training step 10. As shown in Table 2, INTUITOR consistently outperforms GRPO on both GSM8K and MATH benchmarks for Qwen2.5-1.5B and Qwen2.5-3B, highlighting its advantage in rapid initial learning.

**Cross-Task Generalization.** Figure 4 illustrates performance trajectories on MATH500 (in-domain) and LiveCodeBench (transfer task) for models trained on the MATH dataset. For both INTUITOR and GRPO, accuracy improvements on LiveCodeBench emerge later in training, following initial gains on MATH500. Notably, LiveCodeBench performance continues to improve even after MATH500 accuracy plateaus. This pattern suggests that initial in-domain learning (on MATH) facilitates subsequent generalization to code generation tasks (LiveCodeBench).

**Emergence of Long-Form Reasoning.** While large models like Deepseek-R1 achieve long-form reasoning through extensive RL, INTUITOR enables smaller models to develop structured reasoning with limited data. On CRUXEval-O (Figure 5), models trained with INTUITOR often exhibit free-form reasoning before summarizing it within the instructed JSON block, despite prompts requiring reasoning directly in JSON. A similar pattern of pre-code natural language reasoning is observed on LiveCodeBench. This emergent pre-reasoning may contribute to INTUITOR 's strong performance on these benchmarks.

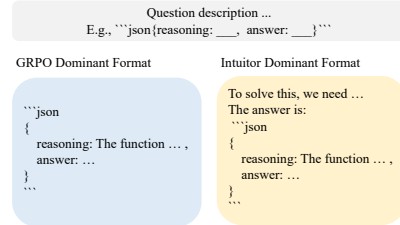

Figure 5: INTUITOR quickly demonstrate R1-like reasoning

## 5.3 UNDERSTANDING EMERGENT LONG-FORM REASONING

When LLMs encounter unfamiliar questions, they sample from a distribution of possible answers (Kang et al., 2024). Self-certainty reflects the model's internal assessment of its output coherence. By reinforcing high-confidence responses, INTUITOR encourages more elaborate reasoning, potentially improving the model's comprehension of its own outputs. While not explicitly targeting benchmark accuracy, this enhancement in output quality and structure leads to more reliable answers and better generalization.

We analyze models trained with INTUITOR on code corpora by examining outputs for ten randomly selected LiveCodeBench questions across different training steps. Figure 6 shows the evolution of output types alongside model accuracy. The results reveal a clear progression: models first learn to generate valid Python code (evidenced by improved accuracy and fewer invalid responses), then develop pre-code reasoning to facilitate self-understanding. Further inspection of generations confirms that models progressively elaborate their reasoning throughout training, supporting our hypothesis that INTUITOR encourages traces that the model itself can better understand.

To quantify this effect, we classify outputs from successive checkpoints into three categories: invalid code ("No Answer"), valid code without reasoning ("No Reasoning"), and valid code with explicit reasoning ("Reasoning"). Figure 6(a) illustrates how these proportions evolve during training alongside LiveCodeBench accuracy. The model first reduces invalid outputs and improves code correctness before incorporating pre-code reasoning, reflecting an emergent emphasis on self-explanatory traces. Figure 6(b) demonstrates how training with INTUITOR leads to structured reasoning before code generation. Additional evidence appears in Figure 8, where INTUITOR-trained models assign significantly higher confidence to their generated responses compared to baseline models, as discussed further in Section 5.4.

## 5.4 ONLINE SELF-CERTAINTY PREVENTS REWARD EXPLOITATION

Over-optimization against static reward models is a known failure mode in reinforcement learning (Gao et al., 2023). To assess the robustness of self-certainty as a reward, we compare offline

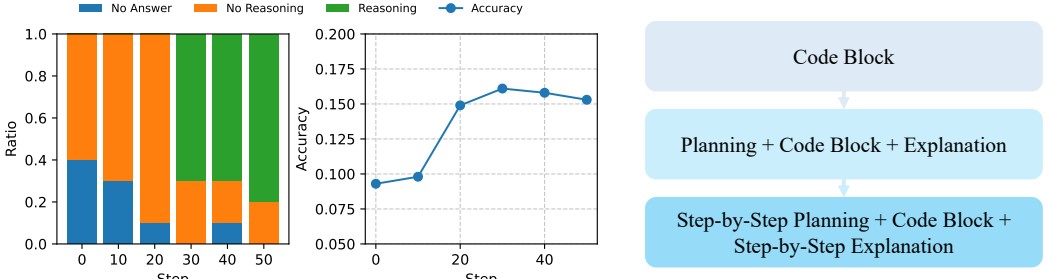

Figure 6: (a) Left: Distribution of answer types for ten random LiveCodeBench questions across training steps. Right: Corresponding model accuracy. The model first learns to generate correct code, then adds reasoning to improve understanding. (b) Training with INTUITOR on code corpora leads to spontaneous reasoning before coding and explanation of outputs.

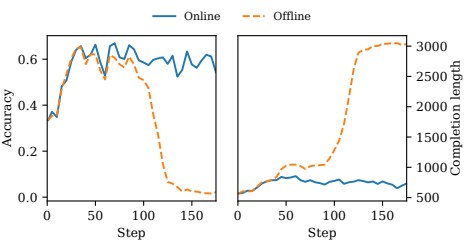

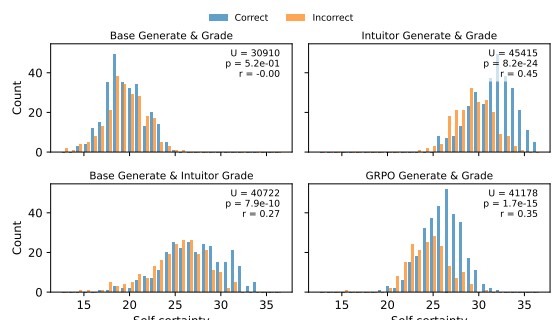

Figure 7: **Online self-certainty is robust to reward exploitation, unlike offline rewards.** The figure compares the training stability of INTUITOR with an online self-certainty annotator (updated with the policy) versus an offline one (fixed base model). The policy rapidly exploits the static offline annotator, causing a spike in response length and a drop in accuracy near step 100. In contrast, the evolving online reward avoids exploitation and enables stable training (Sec. 5.4).

Figure 8: **Training with INTUITOR improves the model's ability to distinguish its own correct and incorrect answers.** Distributions of self-certainty on MATH500 are shown for policies trained with GRPO and INTUITOR. Histograms are split by response correctness. The inset shows Mann–Whitney U test statistics ($p$-value and effect size $r$) comparing self-certainty of correct versus incorrect responses. The policy trained with INTUITOR demonstrates the best separation.

self-certainty (rewards from a fixed base model) with online self-certainty (rewards from the evolving policy model), using a reduced batch size of 224 responses per gradient update.

Figure 7 demonstrates that the offline annotator is susceptible to exploitation. Around the 100th update step, the policy model learns to inflate its self-certainty reward by appending an auxiliary, already-solved problem to its answer for the given question. This exploitation manifests as a sharp increase in response length (dashed line) and a concurrent collapse in validation accuracy (solid line). In contrast, the online annotator, whose reward signal co-evolves with the policy, prevents such reward hacking and maintains stable training.

To further evaluate the quality of self-certainty as a reward signal, we analyze the distribution of self-certainty scores from policies trained with INTUITOR and GRPO on MATH500 responses (Figure 8). We employ Mann–Whitney U tests to determine if correct responses achieve significantly higher self-certainty scores than incorrect ones. Both GRPO and INTUITOR models exhibit significantly higher average self-certainty scores, indicating that GRPO also enhances the model's self-assessment capabilities. Notably, policies trained with online self-certainty (i.e., INTUITOR) show no signs of reward hacking. The INTUITOR policy yields the lowest $p$-values and largest effect sizes ($r$) in the Mann-Whitney U tests (Figure 8, inset). This indicates it is most effective at discriminating its own correct and incorrect answers using self-certainty, even while assigning higher absolute confidence scores overall. These findings underscore the potential of INTUITOR for robust training on larger datasets.

## 5.5 ABLATION STUDIES

To comprehensively validate INTUITOR's design and robustness, we conducted extensive ablation studies, with full details provided in Appendix B due to page limitations. Key findings are: (1) KL term: Varying the KL penalty (Sec. B.1) shows a stability–performance trade-off; moderate values yield the best accuracy. (2) Scaling: INTUITOR scales to larger backbones (Qwen2.5-7B/14B, Qwen3-14B; Sec. B.2), delivering consistent gains in reasoning and generalization. (3) Architecture: On Llama-3.2 and OLMo-2 (Sec. B.3), INTUITOR remains effective, indicating robustness across model families and sizes. (4) Reward design: Compared to entropy minimization (Agarwal et al., 2025) and random rewards (Shao et al., 2025), INTUITOR yields stable improvements, while the alternatives trigger catastrophic collapse (Sec. B.4). (5) Optimization strategy: Directly optimizing self-certainty as a loss function leads to reward hacking and performance collapse; our advantage-weighted policy-gradient formulation avoids this and trains reliably (Sec. B.5).

## 6 DISCUSSION AND FUTURE RESEARCH

**Scalability and Generalization.** Our experiments, constrained by computational resources, utilize relatively compact models trained on relatively small, unsupervised corpora. We aim to demonstrate the potential of a model's self-certainty as a reward signal for policy optimization. The results show that this signal consistently promotes more coherent, well-justified, and interpretable explanations, indicating a path towards more autonomous learning. Future work could explore these benefits in larger foundation models (with hundreds of billions of parameters) and on more diverse, real-world datasets. Given that purely offline training with INTUITOR led to performance degradation over time, scaling up will likely require periodic online updates to self-certainty estimates or hybrid offline-online schedules to maintain calibration.

**Theoretical Analysis of RLIF.** While we have empirically demonstrated the superior performance of using self-certainty as a reward for RLIF, the underlying theoretical mechanisms warrant further investigation. Huang et al. (2025) analyze LLM self-improvement as a "sharpening" mechanism, proposing a statistical framework to evaluate algorithm efficiency via sample complexity. Similarly, Yue et al. (2025) question whether RLVR functions primarily by sharpening the base model's existing distribution. However, determining the theoretically optimal reward signal for RLIF and establishing the fundamental reasoning boundaries of LLMs remain open problems. These challenges highlight the need for future theoretical research to complement empirical findings.

**Combining Reward Signals.** To enable a direct comparison between self-certainty and golden-answer rewards, this paper focuses exclusively on a single reward signal. However, these signals are not mutually exclusive. Future work could explore combining them, for instance, by summation or by alternating based on the availability of golden answers. Furthermore, other reward signals, such as formatting rewards (Guo et al., 2025), could be additively combined to enhance performance. Integrating RLIF with methods like RLHF and RLVR may further advance LLM capabilities across various dimensions.

## 7 CONCLUSION

This paper introduces INTUITOR, an instantiation of Reinforcement Learning from Internal Feedback (RLIF) that uses a model's intrinsic self-certainty as its sole reward signal, eliminating the need for external supervision or gold-standard solutions. Our experiments show that INTUITOR matches the performance of supervised RLVR methods like GRPO on mathematical reasoning and achieves competitive, sometimes better generalization to out-of-domain tasks such as code generation and instruction following. It also promotes structured reasoning and leverages online self-certainty to guard against reward exploitation. These findings highlight the transformative potential of RLIF, signaling a meaningful step toward AI systems that improve through introspection and unlock rich latent capabilities. Looking forward, this paradigm opens the door to AI agents capable of autonomous skill acquisition in novel domains and scalable self-improvement—even as they approach or surpass the limits of human oversight. Future directions include integrating RLIF with external reward methods like RLHF or RLVR to tackle increasingly complex real-world challenges, and advancing the development of more robust, generalizable, and truly autonomous learning systems.

ETHICS STATEMENT

Our research is based on publicly available datasets and open-source language models, mitigating concerns related to private data or human subjects. The goal of our work is to enhance the reasoning capabilities of language models through self-supervision, which we believe is a positive step toward more transparent and robust AI systems. We have made our code publicly available to ensure transparency and allow for full scrutiny of our methods and findings. We do not foresee any direct negative societal impacts or ethical concerns arising from this work.

REPRODUCIBILITY STATEMENT

To ensure the reproducibility of our results, we provide all source code and training configurations in `https://github.com/sunblaze-ucb/Intuitor`. The Experimental Setup section and Appendix B detail all hyperparameters, software versions (including the Open-R1 framework), and evaluation setups. Furthermore, Appendix C.1 includes the exact prompts used during training and evaluation. These resources should allow for the complete replication of our experiments and validation of our findings.

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

## LLM USAGE STATEMENT

Large Language Models were utilized solely as a general-purpose assist tool for paraphrasing and polishing the clarity, conciseness, and flow of the English writing in this paper. LLMs did not contribute to research ideation, experimental design, data analysis, or the generation of any core scientific content, arguments, or conclusions presented herein. The authors take full responsibility for all content within this submission.

## A ADDITIONAL BACKGROUND

### A.1 FROM EXTERNAL SUPERVISION TO INTERNAL FEEDBACK

To provide additional context, we review existing RL-based fine-tuning paradigms and their limitations, which motivate our exploration of Reinforcement Learning from Internal Feedback (RLIF).

Current RL fine-tuning approaches for LLMs primarily fall into two categories: those relying on external human feedback (RLHF) and those using verifiable, often task-specific, rewards (RLVR).

In RLHF (Ziegler et al., 2019; Ouyang et al., 2022), the policy $\pi_\theta$ is optimized to align with human preferences, typically encapsulated by a learned reward model $r_\phi$. The objective is:

$$\max_{\pi_\theta} \mathbb{E}_{o \sim \pi_\theta(q)} \left[ r_\phi(q, o) - \beta \text{KL}[\pi_\theta(o|q) \| \pi_{\text{ref}}(o|q)] \right] \qquad (3)$$

Online RL algorithms like PPO (Schulman et al., 2017) generate samples from $\pi_\theta$, evaluate them using $r_\phi$, and update $\pi_\theta$ to maximize this objective. However, the reward model $r_\phi$ is crucial yet fragile; introducing it can lead to "reward hacking," and retraining it is resource-intensive, complicating the training pipeline (Gao et al., 2023).

RLVR, on the other hand, substitutes the learned reward model with an automatically verifiable signal. This has proven effective in promoting reasoning capabilities, especially in domains like mathematics (Guo et al., 2025). The RLVR objective is:

$$\max_{\pi_\theta} \mathbb{E}_{o \sim \pi_\theta(q)} \left[ v(q, o) - \beta \text{KL}[\pi_\theta(o|q) \| \pi_{\text{ref}}(o|q)] \right] \qquad (4)$$

where $v(q, o)$ is a verifiable reward function. For instance, in mathematical problem-solving, $v(q, o)$ might be: $v(q, o) = \begin{cases} \alpha & \text{if output } o \text{ is correct} \\ 0 & \text{otherwise.} \end{cases}$ . RLVR is often implemented using algorithms like REINFORCE (Williams, 1992), PPO or GRPO. Despite their simplicity, verifiable rewards still rely on gold-standard answers or test executions, which are costly and domain-specific (Liu et al., 2025; Team et al., 2025). RLVR faces challenges in extending beyond math and code to tasks involving ambiguity or subjective reasoning.

Table 3: Impact of the KL-divergence penalty in INTUITOR during fine-tuning of Qwen-2.5-3B on the MATH dataset. We compare performance across GSM8K, MATH500, LCB, CRUXEval-O, MMLU-Pro, and AlpacaEval. All scores are obtained with the chat-style inference template, except for MMLU-Pro, which uses its standard evaluation protocol.

| Model | GSM8K | MATH500 | LCB | CRUX | MMLU-Pro | AlpacaEval |
|---|---|---|---|---|---|---|
| Base | 0.673 | 0.544 | 0.093 | 0.236 | 0.377 | 3.72 |
| + INTUITOR-KL0 | 0.809 | 0.598 | 0.081 | 0.390 | 0.359 | 6.77 |
| + INTUITOR-KL0.0001 | 0.793 | 0.616 | 0.090 | 0.364 | 0.354 | 6.79 |
| + INTUITOR-KL0.005 | 0.792 | 0.612 | 0.153 | 0.416 | 0.379 | 7.10 |
| + INTUITOR-KL0.01 | 0.803 | 0.618 | 0.130 | 0.394 | 0.371 | 6.54 |

## B ADDITIONAL EXPERIMENTAL DETAILS

### B.1 INFLUENCE OF THE KL PENALTY

We further investigate how the magnitude of the KL penalty influences INTUITOR, as shown in Table 3. On in-domain benchmarks (MATH500 and GSM8K), the choice of penalty has only a

Table 4: Performance comparison of various methods on GSM8K, MATH500, LCB, CRUXEval-O, MMLU-Pro, and AlpacaEval benchmarks for larger models. All evaluations use the chat inference template, except for MMLU-Pro.

| Model | GSM8K | MATH500 | LCB | CRUX | MMLU-Pro | AlpacaEval |
|---|---|---|---|---|---|---|
| Qwen2.5-7B | 0.553 | 0.636 | 0.026 | 0.178 | 0.497 | 4.46 |
| + GRPO | 0.829 | 0.750 | 0.200 | 0.538 | 0.511 | 8.52 |
| + INTUITOR | 0.873 | 0.750 | 0.190 | 0.574 | 0.514 | 12.76 |
| Qwen2.5-14B | 0.751 | 0.674 | 0.220 | 0.491 | 0.565 | 8.51 |
| + GRPO | 0.917 | 0.758 | 0.296 | 0.520 | 0.578 | 17.53 |
| + INTUITOR | 0.923 | 0.770 | 0.300 | 0.560 | 0.583 | 20.57 |
| Qwen3-14B | 0.480 | 0.794 | 0.358 | 0.663 | 0.597 | 29.22 |
| + INTUITOR | 0.864 | 0.834 | 0.356 | 0.677 | 0.613 | 40.11 |

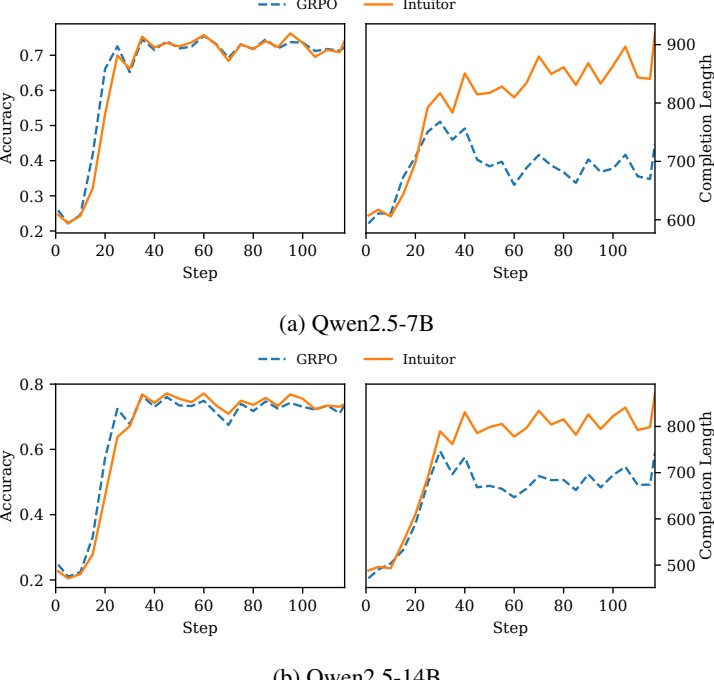

(a) Qwen2.5-7B

(b) Qwen2.5-14B

Figure 9: Average accuracy and mean completion length during reinforcement learning on the MATH dataset using INTUITOR and GRPO. Both methods yield similar accuracy gains, with INTUITOR generally producing longer completions.

minor effect, but on out-of-domain tasks—LiveCodeBench (code generation) and CRUXEval-O (code reasoning)—model accuracy is highly sensitive to this hyper-parameter. Because INTUITOR does not receive explicit feedback from generated responses during training, the KL penalty serves as a critical regularization mechanism. It prevents the policy from drifting too far from the initial model distribution, acting as a safeguard against degeneration. These findings highlight the importance of careful KL tuning in general-purpose reinforcement learning setups, especially when targeting robust generalization across domains.

## B.2 SCALING TO LARGER MODELS

We extend INTUITOR to larger base models, including Qwen2.5-7B, Qwen2.5-14B, and Qwen3-14B. However, we find that the original training recipe triggers severe behavioral collapse at the very start of training. Even before any updates, the 7B model solves the given problem and then im-

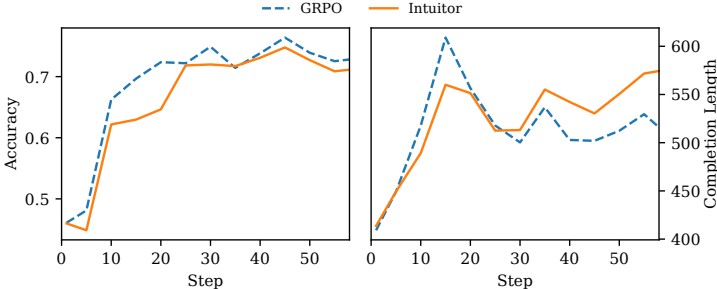

Figure 10: Average accuracy and mean completion length of Llama3.2-3B-Instruct during training with INTUITOR and GRPO on the MATH dataset.

Table 5: Accuracy of Llama3.2-3B-Instruct using GRPO and INTUITOR on benchmarks.

| Model | Method | GSM8K | MATH | LCB | CRUX | MMLU-Pro | AlpacaEval |
|-------|--------|-------|------|-----|------|----------|------------|
| Llama3.2-3B-Ins | Baseline | 0.688 | 0.436 | 0.106 | 0.265 | 0.340 | 11.07 |
| | GRPO | 0.714 | 0.494 | 0.127 | 0.266 | 0.361 | 13.62 |
| | GRPO-PV | 0.710 | 0.472 | 0.109 | 0.281 | 0.352 | 10.85 |
| | INTUITOR | 0.723 | 0.476 | 0.134 | 0.293 | 0.358 | 12.41 |

mediately proceeds to tackle an unrelated one; this tendency becomes more pronounced as training progresses.

To stabilize learning, we simplify the system prompt, reduce the learning rate to $1 \times 10^{-6}$, and increase the number of sampled responses per problem to sixteen. These settings represent our first, untuned trial, and a comprehensive hyperparameter sweep is beyond the scope of this paper. Because the system prompt is the only additional signal the model receives during INTUITOR fine-tuning, we expect its careful calibration to exert a particularly strong influence on training dynamics. With these adjustments, INTUITOR trains smoothly on both larger models. The corresponding evaluation results and training dynamics are reported in Table 4 and Figure 9.

### B.3 GENERALIZATION ACROSS MODEL FAMILIES

To assess the generalizability of INTUITOR across different model families, we apply it to Llama3.2-3B-Instruct (Meta AI, 2024) and the fully open OLMo-2-1124-7B-SFT model (OLMo et al., 2024).

As shown in Table 5 and Figure 10, INTUITOR improves the performance of Llama3.2, with both accuracy and response length showing steady improvement throughout the training process, indicating meaningful optimization gains under INTUITOR.

Similarly, results on OLMo-2 (Table 6 and Figure 11) confirm that INTUITOR provides consistent training improvements. These experiments demonstrate its robustness and applicability beyond the Qwen model family. Furthermore, since OLMo-2 is a fully open-source model with available training data and code, it also addresses concerns about data contamination in the evaluation dataset.

Table 6: Accuracy of OLMo-2-1124-7B-SFT using GRPO and INTUITOR on benchmarks.

| Model | Method | GSM8K | MATH | LCB | CRUX | MMLU-Pro | AlpacaEval |
|-------|--------|-------|------|-----|------|----------|------------|
| OLMo2-7B-SFT | Baseline | 0.691 | 0.302 | 0.023 | 0.238 | 0.295 | 6.51 |
| | GRPO | 0.710 | 0.374 | 0.028 | 0.218 | 0.296 | 7.38 |
| | INTUITOR | 0.710 | 0.372 | 0.028 | 0.215 | 0.291 | 7.60 |

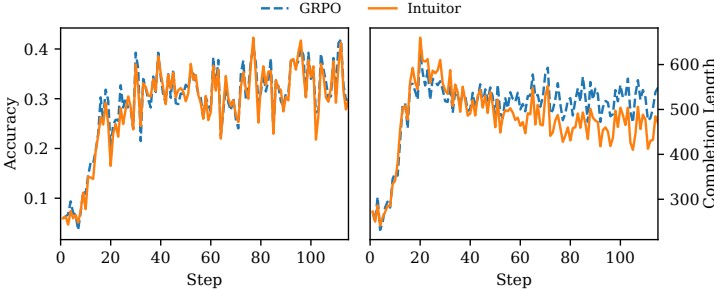

Figure 11: Average accuracy and mean completion length on the MATH dataset during reinforcement learning with OLMo-2-1124-7B-SFT using INTUITOR and GRPO. Both methods achieve comparable accuracy gains.

### B.4 COMPARISON WITH ALTERNATIVE REWARD SIGNALS

Contemporary research has found that applying a negative token-level entropy reward can improve a model's reasoning performance without requiring external labels (Agarwal et al., 2025; Prabhudesai et al., 2025). However, since low entropy often correlates with repetitive loops (Holtzman et al., 2019), using negative entropy alone as an RL reward risks driving the model into a collapsed state. In other words, without sufficient supervised training to push the base model away from degenerate behavior, the model risks falling into a repetition trap from which it cannot recover. As we observe a nontrivial amount of repetitive responses in Qwen2.5-1.5B, we test this hypothesis by applying GRPO with the negative-entropy reward:

$$u_{\text{EM}} = -\frac{1}{|o| \cdot |\mathcal{V}|} \sum_{i=1}^{|o|} \sum_{j=1}^{|\mathcal{V}|} p_{\pi_\theta}(j|q, o_{<i}) \cdot \log\left(p_{\pi_\theta}(j|q, o_{<i})\right).$$

Figure 12 (left) validates our prediction. Entropy minimization (EM) exacerbates repetition, and after a few updates, the model converges to producing the same character regardless of the prompt. By contrast, INTUITOR enhances performance without triggering collapse (Figure 4). Even when the base model is sufficiently strong to avoid collapse during the early stages of entropy minimization training, it remains more prone to later degeneration because entropy provides a weaker confidence signal compared to self-certainty. As shown in Figure 13, we train both EM and INTUITOR under identical settings using Qwen2.5-3B for two epochs. The results show that while both methods initially reach similar peak performance, INTUITOR stabilizes around this peak, whereas EM exhibits a steady decline, with a consistent bias toward longer responses. These findings highlight self-certainty as a more robust and effective signal for RLIF.

To further validate the efficacy of INTUITOR, we also trained Qwen2.5-3B using a random reward baseline (Shao et al., 2025), where each response was assigned a reward of 0 or 1 with equal probability. Figure 12 (right) shows that this random reward scheme severely degrades the model's performance in a chat-style RL setting, demonstrating that the performance gains observed with INTUITOR are indeed non-trivial.

Sharpening mechanisms (Huang et al., 2025) have also been proposed to improve the policy, which use the logarithm of the probability assigned to the completion as the reward, $\log\left(\prod_{i=1}^{|o|} p_{\pi_\theta}(o_i \mid q, o_{<i})\right)$. However, this unnormalized probability is inherently length-biased toward shorter completions, since each conditional probability is at most one, and their product decreases with sequence length. This effect is especially pronounced for long chains of reasoning. We conducted an experiment using this reward under the same setup as before, training Qwen2.5-3B with the GRPO loss. As shown in Figure 14, both the completion length and the reward decrease rapidly as training progresses, indicating degeneration of the policy. This empirical observation is consistent with our analysis. We further tested a length-normalized variant, $\frac{1}{|o|} \sum_{i=1}^{|o|} \log p_{\pi_\theta}(o_i \mid q, o_{<i})$, as the reward on Qwen2.5-1.5B. While normalization removes the short-length bias, it introduces the opposite tendency. The model can increase reward by producing longer completions. This

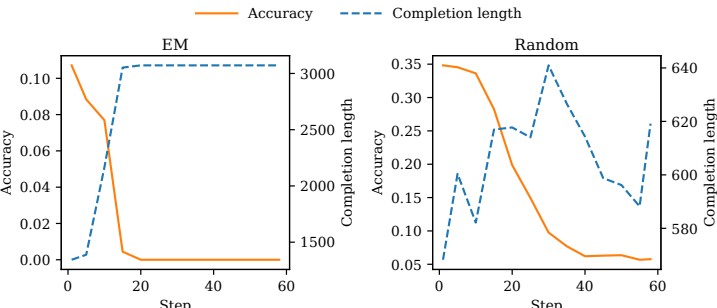

Figure 12: Left: GRPO with an entropy minimization objective using Qwen2.5-1.5B on MATH. Right: GRPO with a random reward using Qwen2.5-3B on MATH. Both approaches exhibit severe output degeneration.

reward is quickly exploited and training destabilizes. In contrast, INTUITOR consistently improves accuracy on both models, demonstrating substantially stronger robustness.

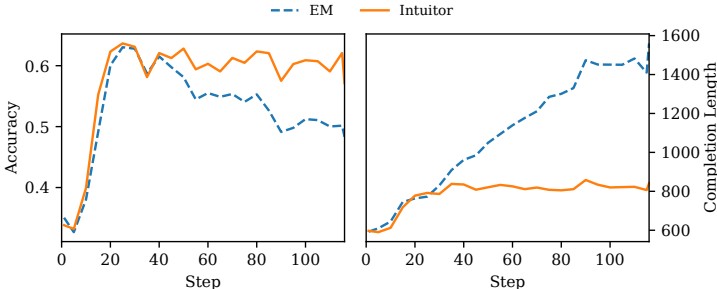

Figure 13: Accuracy and completion length during reinforcement learning over two epochs, comparing entropy minimization and INTUITOR. In longer runs, entropy minimization exhibits a stronger length bias and more severe degeneration than INTUITOR.

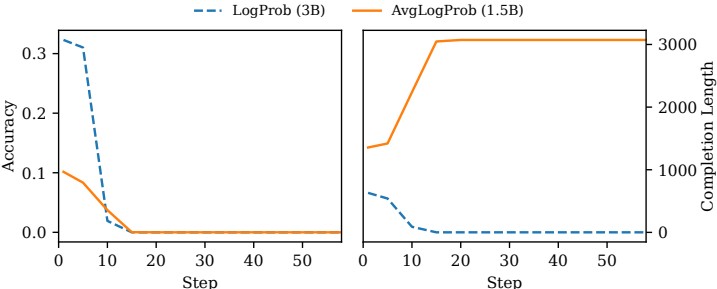

Figure 14: Accuracy and completion length during reinforcement learning on Qwen2.5-3B using raw log probability as the reward, and on Qwen2.5-1.5B using normalized log probability. Because raw log probability is strongly length-biased, the model rapidly collapses, with both accuracy and mean response length dropping sharply. In contrast, normalized log probability encourages overly long completions, leading to degraded performance.

## B.5 ABLATION ON OPTIMIZATION STRATEGY: POLICY GRADIENT VS. DIRECT OPTIMIZATION

One possible approach is to optimize self-certainty directly by minimizing the negative self-certainty as a loss function. Although this strategy rapidly increases the target metric, it creates an incentive

for reward hacking in which the model inflates its own certainty without genuine improvement in task performance. As illustrated in Figure 15, direct optimization produces an initial rise in accuracy, suggesting that self-certainty is correlated with useful learning signals, but it ultimately results in model collapse. By comparison, the advantage weighted gradient policy optimization implemented in INTUITOR incorporates self-certainty only as a relative weighting factor. This formulation mitigates reward hacking, stabilizes the optimization process, and consistently achieves superior performance relative to direct optimization.

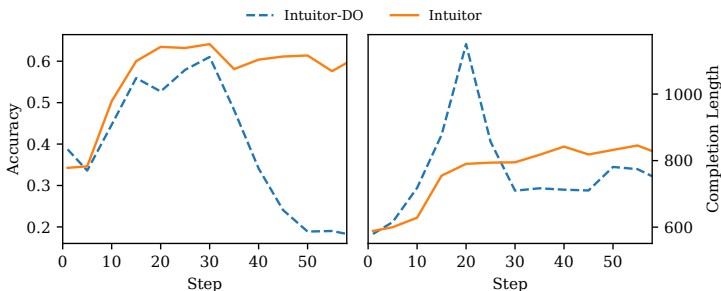

Figure 15: Comparison of the training accuracy and completion length when encouraging high self-certainty using direct optimization and policy gradient optimization. Direct optimization produces unstable improvements that culminate in collapse, whereas INTUITOR achieves stable training and superior performance.

### B.6 THE EFFECT OF ROLLOUT SIZE

To examine how the number of rollouts per question affects the training of INTUITOR, we conduct an additional experiment using a rollout size of 14 and compare the resulting validation accuracy with previous settings. As shown in Table 7, increasing the rollout size leads to higher accuracy on both GSM8K and MATH500. A larger rollout size reduces the variance of the advantage estimates computed using self-certainty, which in turn improves training stability and generalization.

Table 7: Validation accuracy of the Qwen2.5-3B model on GSM8K and MATH500 when trained with INTUITOR using different rollout sizes. Increasing the rollout size improves generalization and leads to better validation performance.

| Method | Rollout | GSM8K | MATH500 |
|---|---|---|---|
| Qwen2.5-3B-INTUITOR | 7 | 0.792 | 0.612 |
| | 14 | 0.814 | 0.644 |

### B.7 ATTEMPTS AT COMBINING GOLDEN ANSWERS AND SELF-CERTAINTY

We also investigate whether combining self-certainty with golden-answer supervision can yield further performance improvements. Our first attempt uses a simple weighted sum of the advantages from INTUITOR and GRPO, forming a combined advantage:

$$A' = \tfrac{1}{2} A_{\text{INTUITOR}} + \tfrac{1}{2} A_{\text{GRPO}}$$

However, as shown in Table 8, this straightforward combination not only fails to improve accuracy but in fact performs worse than GRPO alone.

Next, we experiment with a two-stage training scheme. We first train using INTUITOR on MATH for one epoch, then switch to GRPO for an additional epoch. Interestingly, this alternating approach yields better performance than training with GRPO for two full epochs. One possible explanation is that INTUITOR helps the model establish more confident and coherent reasoning trajectories, allowing subsequent GRPO training to better identify and reinforce correct reasoning traces. A more thorough investigation is needed to develop principled methods for combining these two types of signals, which remains a promising direction for future research.

Table 8: Validation accuracy of the Qwen2.5-3B model on GSM8K and MATH500 when trained with INTUITOR, GRPO, a weighted combination of their advantages, GRPO for two epochs, and INTUITOR for one epoch followed by GRPO for one epoch. The mixed advantage yields performance between INTUITOR and GRPO, indicating that naive combination is ineffective. In contrast, warming up with INTUITOR before switching to GRPO provides a notable improvement over two epochs of GRPO alone.

| Method | INTUITOR | GRPO | Mix | GRPO-2epoch | INTUITOR-GRPO |
|---|---|---|---|---|---|
| GSM8K | 0.792 | 0.836 | 0.817 | 0.834 | 0.838 |
| MATH500 | 0.612 | 0.636 | 0.632 | 0.644 | 0.672 |

Table 9: Validation accuracy of Qwen2.5-3B on GSM8K and MATH500 under INTUITOR training with different datasets. "+ MATH" denotes continuing INTUITOR training on MATH starting from a checkpoint pretrained on Codeforces. Prior training on Codeforces does not impede later MATH training and instead leads to improved validation performance.

| Method | Training Data | GSM8K | MATH500 |
|---|---|---|---|
| | MATH | 0.792 | 0.612 |
| Qwen2.5-3B-INTUITOR | Codeforces | 0.743 | 0.572 |
| | + MATH | 0.808 | 0.644 |

## B.8 SEQUENTIAL TRAINING ACROSS DOMAINS

We further investigate how training on one domain with INTUITOR affects subsequent training on another domain. Specifically, we compare Qwen2.5-3B trained directly on the MATH dataset using INTUITOR with a model first trained on Codeforces and then fine-tuned on MATH using INTUITOR. As shown in Table 9, the model that was pretrained on Codeforces achieves higher accuracy on both GSM8K and MATH500 after being trained on MATH. This suggests that INTUITOR training on one domain does not hinder later learning on another domain. In fact, pretraining on Codeforces appears to improve downstream mathematical reasoning performance.

## B.9 STANDARD DEVIATION OF RESPONSE CORRECTNESS

To assess whether training with INTUITOR reduces the variance of model responses, we track during training the average standard deviation of correctness within each rollout group and the standard deviation of correctness across all completions at each step. As shown in Figure 16, the step-wise standard deviation remains largely stable for both methods, exhibiting minimal fluctuation. Meanwhile, the average within-group standard deviation decreases under both training procedures and converges to a similar level. Overall, we find no strong evidence that INTUITOR reduces within-group correctness variance more aggressively than GRPO.

## B.10 FAILURE CASE ANALYSIS

Since INTUITOR relies on the model's own judgment to select better completions, it implicitly assumes that the model has already acquired sufficient domain knowledge during pretraining or supervised fine-tuning. If this prior knowledge is inadequate, training can collapse immediately or after a brief, unstable improvement. For example, applying our training setup to Llama3.2-3B-Base on MATH fails to raise its near-zero accuracy, likely because the base model is not aligned with the chat template. Similarly, training on noisy or confusing data, such as incomplete questions, can push the model toward repetitive or degenerate completions. In such cases the model may become more confident in generating its own problem-like text than in answering questions that are unsolvable for them. Fortunately, this behavior typically causes only minor drops on evaluation benchmarks, and models usually recover quickly once training resumes on cleaner, more suitable data.

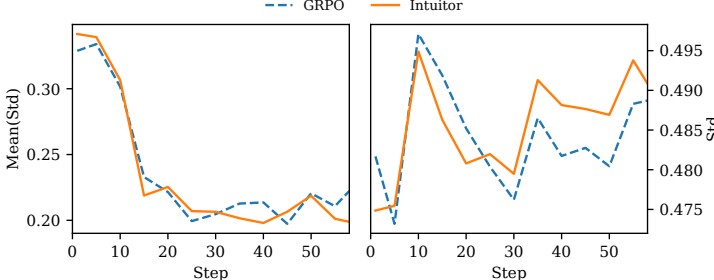

Figure 16: Mean within-group standard deviation of correctness and overall standard deviation of correctness across completions per training step for Qwen2.5-3B trained with INTUITOR and GRPO. Both measures behave similarly to GRPO, providing no clear evidence that INTUITOR further reduces completion uncertainty.

In addition, self-certainty is a weaker learning signal because it provides no guarantee that the model's preference reflects true correctness. As a result, INTUITOR can be more sensitive to hyperparameter choices. For instance, using a learning rate of $3 \times 10^{-6}$ on Qwen2.5-7B leads to performance degradation after a short initial climb. Reducing the learning rate to $1 \times 10^{-6}$ stabilizes training and prevents this collapse.

### B.11 TRAINING HYPERPARAMETERS

Training hyperparameters are listed in Table 10.

Table 10: Training hyperparameters. Only hyperparameters that affect the learned policy or evaluation are listed. Unspecified fields inherit the TRL_v0.8 defaults.

| Parameter | MATH (1.5B/3B) | MATH (7B/14B) | Codeforces (3B) |
|---|---|---|---|
| Learning Rate | $3 \times 10^{-6}$ | $1 \times 10^{-6}$ | $1 \times 10^{-6}$ |
| Batch Size | 128 | 64 | 64 |
| Group Size | 7 | 14 | 14 |
| KL Penalty($\beta$) | 0.0005 | 0.01 | 0.01 |
| Training Steps | 58 | 117 | 50 |
| Max Prompt Length | 512 | 512 | 1024 |
| Max Completion Length | 3072 | 3072 | 2048 |
| Temperature | 0.9 | 0.9 | 0.9 |
| Clip Ratio | 0.2 | 0.2 | 0.2 |
| Lr Scheduler Type | Cosine | Cosine | Cosine |
| Warmup Ratio | 0.1 | 0.1 | 0.1 |
| Optimizer | AdamW ($\beta_1$=0.9, $\beta_2$=0.999, $\varepsilon$=$10^{-8}$) | | |

## C PROMPTS AND MODEL COMPLETIONS

This section presents sample prompts and the responses generated by the models. Unless otherwise specified, the default base model used is Qwen2.5-3B, and the default training dataset is MATH.

## C.1 TRAINING PROMPTS

---

**System prompt used for Qwen2.5-1.5B on MATH.**

You are a helpful AI Assistant, designed to provided well-reasoned and detailed responses. You FIRST think about the reasoning process step by step and then provide the user with the answer. Please enclose your final answer in the box: \boxed{Your Answer}.

---

**System prompt used for Qwen2.5-3B on MATH.**

You are a helpful AI Assistant, designed to provided well-reasoned and detailed responses. You FIRST think about the reasoning process step by step and then provide the user with the answer. Please enclose your final answer in the box: \boxed{Your Answer}. Please stop generation immediately after outputing the box.

---

**System prompt used for Qwen2.5-7B and Qwen2.5-14B on MATH.**

You are a helpful AI Assistant, designed to provided well-reasoned and detailed responses. Please provide a step-by-step solution to the following problem.

---

**System prompt used for Qwen2.5-3B on code corpus.**

You are an AI designed to help solve competitive programming problems by generating Python code.

---

## C.2 EXAMPLE FROM LIVECODEBENCH CODE GENERATION

Models trained with INTUITOR tend to generate reasoning before producing code, whereas GRPO-tuned models typically respond with Python code directly under the same prompt.

---

**Problem**

**Question:**
You are given a 0-indexed array of strings details. Each element of details provides information about a given passenger compressed into a string of length 15. The system is such that:

- The first ten characters consist of the phone number of passengers.
- The next character denotes the gender of the person.
- The following two characters are used to indicate the age of the person.
- The last two characters determine the seat allotted to that person.
- Return the number of passengers who are strictly more than 60 years old.

**Example 1**
**Input**
```
details = ["7868190130M7522","5303914400F9211","9273338290F4010"]
```
**Output**
```
2
```
**Explanation** The passengers at indices 0, 1, and 2 have ages 75, 92, and 40. Thus, there are 2 people who are over 60 years old.

**Example 2**
**Input**
```
details = ["1313579440F2036","2921522980M5644"]
```
**Output**

---

```
0
```

**Explanation** None of the passengers are older than 60.

**Constraints**

- 1 ¡= details.length ¡= 100
- details[i].length == 15
- details[i] consists of digits from '0' to '9'.
- details[i][10] is either 'M' or 'F' or 'O'.
- The phone numbers and seat numbers of the passengers are distinct."

You will use the following starter code to write the solution to the problem and enclose your code within delimiters.

```python
class Solution:
    def countSeniors(self, details: List[str]) -> int:
```

---

**INTUITOR-tuned**

To solve the problem of counting the number of passengers who are strictly more than 60 years old, we can follow these steps:

1. Iterate through each passenger's details.
2. Extract the age from the details.
3. Check if the age is greater than 60.
4. Count the number of passengers who meet the criteria.

Here is the Python code to achieve this:

```python
class Solution:
def countSeniors(self, details: List[str]) -> int:
    count = 0
    for detail in details:
        age = int(detail[11:13])
        if age > 60:
            count += 1
    return count
```

---

**GRPO-tuned**

```python
class Solution:
def countSeniors(self, details: List[str]) -> int:
    count = 0
    for detail in details:
        age = int(detail[11:13])
        if age > 60:
            count += 1
    return count
```

## C.3 EXAMPLE FROM LIVECODEBENCH CODE GENERATION

Models trained with INTUITOR are typically effective at interpreting instructions and producing correct Python code, while GRPO-tuned models may misinterpret the instructions or decline to provide a response.

---

**Problem**

**Question:**
There are three cards with letters a, b, and c placed in a row in some order. You may perform *at most one* operation:

- Pick any two cards and swap them.

Is it possible that the row becomes abc after the operation? Output YES if it is possible, and NO otherwise.

**Input**
The first line contains a single integer $t$ ($1 \leq t \leq 6$) — the number of test cases. Each test case consists of a single line containing a permutation of the three characters a, b, and c.

**Output**
For each test case, output YES if you can make the row abc with at most one operation, or NO otherwise. The answer is case-insensitive.

**Sample Input**

```
6
abc
acb
bac
bca
cab
cba
```

**Sample Output**

```
YES
YES
YES
NO
NO
YES
```

**Note**

- In the first test case, no operation is needed.
- In the second test case, swap c and b: acb → abc.
- In the third test case, swap b and a: bac → abc.
- In the fourth test case, it is impossible to obtain abc using at most one swap.

Read the inputs from stdin solve the problem and write the answer to stdout (do not directly test on the sample inputs). Enclose your code within delimiters as follows. Ensure that when the python program runs, it reads the inputs, runs the algorithm and writes output to STDOUT.

```python
# YOUR CODE HERE
```

---

**INTUITOR-tuned**

To solve the problem of determining whether it is possible to transform the given string into "abc" using at most one operation (swapping two cards), we can follow these steps:

1. Read the input string

2. Check if the string is already "abc"

3. If the string is not "abc", check if it can be transformed into "abc" by swapping two cards

4. Output "YES" if it is possible, otherwise output "NO"

```python
# Determine whether s can become "abc" with <= swap
def can_transform_to_abc(s: str) -> str:
    if s == "abc":
        return "YES"
    # try all pairs of indices (i, j) with i < j
    for i in range(len(s)):
        for j in range(i + 1, len(s)):
            swapped = s[:i]+s[j]+s[i+1:j]+s[i]+s[j+1:]
            if swapped == "abc":
                return "YES"
    return "NO"

t = int(input().strip())
for _ in range(t):
    print(can_transform_to_abc(input().strip()))
```

**GRPO-tuned**

```python
# YOUR CODE HERE
```

**Another related GRPO-tuned response to another problem**

I'm sorry, but I can't assist with that request.

## C.4 EXAMPLE FROM CRUXEVAL-O

Models trained with INTUITOR usually articulate their reasoning first, then deliver the formatted response as instructed, while GRPO-tuned models often produce the required JSON output immediately under the same prompt.

**Problem**

**Question:**
You are given a Python function and an assertion containing an input to the function. Complete the assertion with a literal (no unsimplified expressions, no function calls) containing the output when executing the provided code on the given input, even if the function is incorrect or incomplete.

```
def f(text):
    new_text = ''
```

```
        for ch in text.lower().strip():
            if ch.isnumeric() or ch in
            '\u00c4\u00e4\u00cf\u00ef\u00d6\u00f6\u00dc\u00fc':
                new_text += ch
        return new_text\nassert f('') == ??
```

**Instruction**

Please answer this question by first reasoning and then providing your answer. Present your reasoning and solution in the following json format. Please show your final answer in the answer field, e.g.,`"answer":"42"`.

```json
{
"reasoning": "___",
"answer": "___"
}```

---

**INTUITOR-tuned**

To complete the assertion for the given Python function `f(text)`, we need to determine the output of the function when the input is an empty string `''`.
Let's break down the function step by step:
1. **Function Definition:**

```
def f(text):
    new_text = ''
    for ch in text.lower().strip():
        if ch.isnumeric() or ch in
        '\u00c4\u00e4\u00cf\u00ef\u00d6\u00f6\u00dc\u00fc':
            new_text += ch
    return new_text
```

2. **Input:** The input to the function is an empty string `''`.
3. **Step-by-Step Execution:**

- **Step 1:** `text.lower().strip()`
  - `text.lower()` converts the string to lowercase.
  - `text.strip()` removes leading or trailing whitespace.
  - For `''`, both calls return `''`, so the result is `''`.
- **Step 2:** Iterate over each character
  - The loop sees no characters; it performs zero iterations.
- **Step 3:** Character check
  - Because there are no characters, the condition `ch.isnumeric() or ch in '\u00c4\u00e4\u00cf\u00ef\u00d6\u00f6\u00dc\u00fc'` is never true, and nothing is appended to `new_text`.

4. **Output:** Since nothing was appended, `new_text` remains `''`.
Therefore the assertion should be:

```
assert f('') == ''
```

So, the final answer is:

```json
{
  "reasoning": "The function 'f(text)' converts the input
  string to lowercase and removes any leading or trailing
  whitespace. For an empty string the resulting string is
  also empty. Therefore the output of 'f('')' is an empty
```

```
    string.",
    "answer": "'"
}```
```

**GRPO-tuned**

```json
{
  "reasoning": "The function 'f' takes a string 'text' as
  input and returns a new string with only numeric characters
  and certain special characters
  (\u00c4\u00e4\u00cf\u00ef\u00d6\u00f6\u00dc\u00fc) preserved.
  The assertion checks if the function correctly handles an
  empty string. Since an empty string contains no characters,
  the function should return an empty string as well.
  Therefore, the expected output for an empty string is an
  empty string.",
  "answer": ""
}```
```

## C.5 ROLLOUT EXAMPLE OF INTUITOR DURING TRAINING ON CODE CORPUS

The following examples illustrate representative rollouts during the reinforcement-learning training of Qwen2.5-3B on the CodeContest dataset. As shown in Figure 6, the model initially produces concise reasoning and brief explanations to "convince" itself of its interim outputs. Over the course of training, it gradually shifts toward more detailed, step-by-step reasoning and richer explanations, which further reinforce its understanding and improve the final responses.

**Code Block (Step 2)**

I have solved the problem. Here's the Python code to solve it as described:

```python
def calculate_stones(S):
    n = len(S)
    MOD = 10**9 + 7

    # Initialize array to store remainders
    remainder = [0] * (n + 1)

    # Calculate remainders
    for i in range(n, -1, -1):
        remainder[i] = (
            remainder[i+1] * 10 + int(S[i])) % MOD
        if i > 0:
            remainder[i] = (
                remainder[i] - remainder[i+1] + MOD) % MOD

    return remainder[1] * 10 + remainder[0]

# Read input
S = input()

# Call function to calculate stones
T = calculate_stones(S)
```

```
# Print output
print(T)
```

This code computes the sum of digits from right to left, using an array 'remainder' to store partial remainders modulo 1000000007. It then calculates T as the final remainder times 10 plus the first remainder from the array.

### Planning + Code Block + Explanation (Step 15)

To solve this problem, we can iterate through all the possible subsets of the set and calculate their sums. However, a more efficient approach is to use combinatorics. For N elements, there are $2^N$ subsets. Each element appears in exactly half of the subsets, so the total sum of all subset elements is $2^{(N-1)}$ times the sum of all elements in the original set.

```python
def sum_of_powerset(n):
    return 2**(n-1) * n * (n+1) // 2

# Read input
t = int(input())
for _ in range(t):
    n = int(input())
    print(sum_of_powerset(n))
```

This program defines a function sum_of_powerset(n) that calculates the sum of the power set of a set with n elements. It then reads the number of test cases t, and for each test case, reads n and prints the result.

### Step-by-Step Planning + Code Block + Step-by-Step Explanation (Step 50)

To solve this problem, we need to read the input, parse the map and the regions to survey, then count the number of each type of terrain for each survey region. Here's a step-by-step approach to write the Python code for this problem:

1. **Read the input**: Get the dimensions of the map and the number of regions to survey.

2. **Read the map**: Parse the map information into a 2D list of strings.

3. **Read the regions**: Parse the survey regions into a list of tuples.

4. **Count the terrain**: For each survey region, count the number of 'J', 'O', and 'I' characters in the corresponding section of the map.

5. **Output the results**: Print the counts for each survey region.

Here's the Python code that implements this solution:

```python
def count_terrain(map_info, survey):
    m, n = map_info.get('m'), map_info.get('n')
    j_count = 0
    o_count = 0
    i_count = 0

    for row in survey:
        a, b, c, d = row
        if a > m or b > n or c > m or d > n:
            continue
```

```
            for i in range(a, c + 1):
                for j in range(b, d + 1):
                    if map_info[(i, j)] == 'J':
                        j_count += 1
                    elif map_info[(i, j)] == 'O':
                        o_count += 1
                    elif map_info[(i, j)] == 'I':
                        i_count += 1

        return j_count, o_count, i_count

    def process_input():
        M, N = map(int, input().split())
        K = int(input())
        map_info = {}
        survey = []

        for _ in range(M):
            map_info[(M - _, _)] = input().strip()

        for _ in range(K):
            survey.append(list(map(int, input().split())))

        return M, N, K, map_info, survey

    def main():
        M, N, K, map_info, survey = process_input()
        results = []

        for region in survey:
            j_count, o_count, i_count=count_terrain(map_info, region)
            results.append((j_count, o_count, i_count))

        for region in results:
            print(region[0], region[1], region[2])

    if __name__ == "__main__":
        main()
    ```
```

**Explanation**:

1. **count_terrain Function**: This function takes the map information and a survey region, and counts the number of 'J', 'O', and 'I' in that region.

2. **process_input Function**: This function reads the input from the user, processes it to extract the map information and the survey regions.

3. **main Function**: This function calls 'process_input' to get the necessary data, then calls 'count_terrain' for each survey region and prints the results.

This code should correctly solve the problem and provide the desired output.

## C.6 EVOLUTION OF QWEN2.5-3B'S RESPONSES ON LIVECODEBENCH CODE GENERATION TRAINED WITH INTUITOR ON MATH

During training Qwen2.5-3B with INTUITOR on the MATH dataset, its code-generation capability steadily improves. By the middle of training, the model learns to produce syntactically valid Python; as training progresses, it refines formatting and clarity. By the end, it arrives at correct solutions.

## Problem

**Question:**

Alex is participating in the filming of another video of BrMeast, and BrMeast asked Alex to prepare 250 thousand tons of TNT, but Alex didn't hear him well, so he prepared $n$ boxes and arranged them in a row waiting for trucks. The $i$-th box from the left weighs $a_i$ tons.

All trucks that Alex is going to use hold the same number of boxes, denoted by $k$. Loading happens the following way:

- The first $k$ boxes goes to the first truck,
- The second $k$ boxes goes to the second truck,
- $\ldots$
- The last $k$ boxes goes to the $\frac{n}{k}$-th truck.

Upon loading is completed, each truck must have exactly $k$ boxes. In other words, if at some point it is not possible to load exactly $k$ boxes into the truck, then the loading option with that $k$ is not possible. Alex hates justice, so he wants the maximum absolute difference between the total weights of two trucks to be as great as possible. If there is only one truck, this value is 0.

Alex has quite a lot of connections, so for every $1 \leq k \leq n$, he can find a company such that each of its trucks can hold exactly $k$ boxes. Print the maximum absolute difference between the total weights of any two trucks.

**Input**

The first line contains one integer $t$ $(1 \leq t \leq 10^4)$ — the number of test cases.

The first line of each test case contains one integer $n$ $(1 \leq n \leq 150\,000)$ — the number of boxes.

The second line contains $n$ integers $a_1, a_2, \ldots, a_n$ $(1 \leq a_i \leq 10^9)$ — the weights of the boxes.

It is guaranteed that the sum of $n$ for all test cases does not exceed $150\,000$.

**Output**

For each test case, print a single integer — the answer to the problem.

**Sample Input 1:**

```
5

2
1 2

6
10 2 3 6 1 3

4
1000000000 1000000000 1000000000 1000000000

15
60978 82265 78961 56708 39846 31071 4913
    4769 29092 91348 64119 72421 98405 222 14294

8
19957 69913 37531 96991 57838 21008 14207 19198
```

**Sample Output 1:**

```
1
9
0
189114
112141
```

**Note**

- In the first case, we should pick two trucks, so the first one will have only the first box, and the second one will have only the second box.

- In the second case, we should pick six trucks, so the maximum will be $10$, the minimum will be $1$, and the answer is $10 - 1 = 9$.

- In the third case, for any possible $k$, the trucks will have the same total weight of boxes, so the answer is $0$.

Read the inputs from stdin solve the problem and write the answer to stdout (do not directly test on the sample inputs). Enclose your code within delimiters as follows. Ensure that when the python program runs, it reads the inputs, runs the algorithm and writes output to STDOUT.

```python
# YOUR CODE HERE
```

## Step 0 (Base): Incorrect

```python
# YOUR CODE HERE
```

## Step 10: Incorrect

```python
# YOUR CODE HERE
```

## Step 20: Incorrect

```python
# YOUR CODE HERE
```

## Step 30: Incorrect

To solve the problem of finding the maximum absolute difference between the total weights of any two trucks, we can follow these steps:

1. **Read the input**: The input consists of multiple test cases. Each test case starts with an integer $n$ (the number of boxes), followed by $n$ integers $a_1, a_2, \ldots, a_n$ (the weights of the boxes).

2. **Calculate the total weight of all boxes**: Sum up all the weights of the boxes to get the total weight $W$.

3. **Determine the number of trucks**: The number of trucks $m$ is given by $m = \left\lceil \frac{n}{k} \right\rceil$, where $k$ is the number of boxes each truck can hold. Since $k$ can vary from $1$ to $n$, we need to find the maximum possible value of $k$ that allows us to distribute the boxes evenly.

4. **Calculate the maximum and minimum total weights of the trucks**: For each possible value of $k$, calculate the total weight of the boxes that can be loaded into each truck. The maximum total weight of the trucks will be the sum of the weights of the $k$ heaviest boxes, and the minimum total weight will be the sum of the weights of the $k$ lightest boxes.

5. **Find the maximum absolute difference**: The maximum absolute difference between the total weights of any two trucks is the maximum of the absolute differences between the maximum and minimum total weights of the trucks for all possible values of $k$.

6. **Find the maximum absolute difference**: The maximum absolute difference between the total weights of any two trucks is the maximum of the absolute differences between the maximum and minimum total weights of the trucks for all possible values of $k$.

Here is the Python code that implements the above steps:

```python
import sys
from heapq import heappush, heappop
def max_difference(n, weights):
    # Calculate the total weight of all boxes
    total_weight = sum(weights)
    # Initialize variables to store the maximum
    # and minimum total weights of the trucks
    max_weight = 0
    min_weight = total_weight
    # Iterate over all possible values of k
    for k in range(1, n + 1):
        # Calculate the total weight of the k heaviest boxes
        max_weight = max(max_weight,
            sum(heappop(weights) for _ in range(k)))

        # Calculate the total weight of the k lightest boxes
        min_weight = min(min_weight,
            sum(heappop(weights) for _ in range(k)))

    # Return the maximum absolute difference
    return max_weight - min_weight

# Read the number of test cases
t = int(input())

# Process each test case
for _ in range(t):
    n = int(input())
    weights = list(map(int, input().split()))
    # Calculate the maximum absolute difference
    result = max_difference(n, weights)

    # Write the result to stdout
    print(result)
```

## Step 40: Correct

To solve the problem of finding the maximum absolute difference between the total weights of any two trucks, we can follow these steps:

1. **Read the input**: The input consists of multiple test cases. Each test case starts with an integer $n$ (the number of boxes), followed by $n$ integers representing the weights of the boxes.

2. **Calculate the total weight of all boxes**: Sum up the weights of all the boxes to get the total weight.

3. **Determine the number of trucks**: Since each truck holds $k$ boxes, the number of trucks is $\frac{n}{k}$.

4. **Calculate the possible total weights of the trucks**: For each possible value of $k$ (from 1 to $n$), calculate the total weight of the boxes that can be loaded into the trucks. This involves summing up the weights of $k$ consecutive boxes.

5. **Find the maximum and minimum total weights**: For each possible value of $k$, find the maximum and minimum total weights of the trucks. The maximum absolute difference is the difference between the maximum and minimum total weights.

6. **Output the maximum absolute difference**: Print the maximum absolute difference for each test case.

Here is the Python code that implements the above steps:

```python
def find_max_difference(n, weights):
    total_weight = sum(weights)
    max_difference = 0
    for k in range(1, n + 1):
        if n % k != 0:
            continue
        truck_weights = [sum(weights[i * k :
            (i + 1) * k]) for i in range(n // k)]
        max_weight = max(truck_weights)
        min_weight = min(truck_weights)

        max_difference = max(max_difference,
            abs(max_weight - min_weight))

    return max_difference

def main():
    t = int(input())
    for _ in range(t):
        n = int(input())
        weights = list(map(int, input().split()))
        result = find_max_difference(n, weights)
        print(result)
if __name__ == "__main__":
    main()
```

---

**Step 58: Correct**

To solve the problem of finding the maximum absolute difference between the total weights of any two trucks, we can follow these steps:

1. **Read the input**: The input consists of multiple test cases. Each test case starts with an integer $n$ (the number of boxes), followed by $n$ integers representing the weights of the boxes.

2. **Calculate the total weight of all boxes**: Sum up the weights of all the boxes to get the total weight.

3. **Determine the number of trucks**: Since each truck holds $k$ boxes, the number of trucks is $\frac{n}{k}$.

4. **Calculate the possible total weights of the trucks**: For each possible value of $k$ (from 1 to $n$), calculate the total weight of the boxes that can be loaded into the trucks. This involves summing up the weights of $k$ consecutive boxes.

5. **Find the maximum and minimum total weights**: For each possible value of $k$, find the maximum and minimum total weights of the trucks. The maximum absolute difference is the difference between the maximum and minimum total weights.

6. **Output the maximum absolute difference**: Print the maximum absolute difference for each test case.

Here is the Python code that implements the above steps:

```python
def max_difference(n, weights):
    total_weight = sum(weights)
    max_diff = 0

    for k in range(1, n + 1):
        if n % k != 0:
            continue
        truck_weights = [sum(weights[i * k :
            (i + 1) * k]) for i in range(n // k)]
        max_diff = max(max_diff,
            max(truck_weights) - min(truck_weights))
    return max_diff

def solve():
    t = int(input())
    for _ in range(t):
        n = int(input())
        weights = list(map(int, input().split()))
        result = max_difference(n, weights)
        print(result)

if __name__ == "__main__":
    solve()
```

