# OpenReview forum: "Learning to Reason without External Rewards"
_ICLR.cc/2026/Conference — ICLR 2026 Poster_

### Official Review · Reviewer_gp3y · 2025-10-16

**Soundness:** 3
**Presentation:** 3
**Contribution:** 3
**Rating:** 6
**Confidence:** 4

**Summary:**

This paper proposes Reinforcement Learning from Internal Feedback (RLIF) as a paradigm for improving reasoning ability in large language models (LLMs) without relying on external, verifiable rewards. The authors introduce Intuitor, an instantiation of RLIF that replaces the external correctness signal used in methods like GRPO with an intrinsic “self-certainty” metric. Self-certainty is operationalized via the KL divergence between the model’s output distribution and a reference distribution, encouraging more confident and consistent reasoning. The authors demonstrate that Intuitor achieves comparable performance to GRPO on mathematical reasoning benchmarks and shows competitive generalization to out-of-distribution tasks such as code generation. Interestingly, models trained with Intuitor tend to produce longer, more detailed chain-of-thought (CoT) responses, which the authors interpret as reinforcing stepwise reasoning coherence.

**Strengths:**

The paper tackles a timely and important question: how can LLMs learn to reason in domains where explicit supervision or reward verification is infeasible? The proposed self-certainty–based RL framework is elegant, computationally tractable, and broadly applicable.

- The conceptual simplicity of optimising for internal confidence instead of external correctness makes the method appealing for scaling reasoning improvement without costly reward models or test suites.
- The inclusion of multiple model families in the appendix (beyond Qwen) demonstrates the generality of the approach, though this could be highlighted more clearly in the main text.
- The observation that RLIF-trained models produce longer and more detailed CoTs is particularly interesting. The authors’ explanation—that longer CoTs stabilize reasoning trajectories—is plausible and worth further study.

Overall, the paper contributes to the growing discourse on methods for reasoning elicitation in LLMs.

**Weaknesses:**

While the approach is novel and promising, several aspects of the evaluation and analysis could be strengthened.
- The evaluation scope is somewhat narrow. Although the method is positioned as applicable to datasets without verifiable answers, all reported experiments are on domains (math, code) where correctness can in fact be automatically checked. Demonstrating effectiveness on tasks without clear reward signals would make the paper more convincing.
- The mechanistic explanation of how optimizing for self-certainty leads to qualitatively better reasoning remains underspecified. The KL-divergence–based objective effectively sharpens the output distribution, similar to reducing the sampling temperature. It would be helpful for the authors to clarify why RL training with self-certainty yields higher-quality reasoning compared to simply sampling greedily or using low-temperature decoding. It would be helpful to understand in abstract, mathematical terms what is the optimal solution of RL training with self-consistency vs. a greedy sample.
- Claims of superior out-of-distribution generalisation over GRPO appear overstated. For instance, in the MATH→LCB/CRUX transfer setup, the performance gains are small or inconsistent across models (e.g., Qwen2.5-1.5B and OLMo2-7B-SFT show slightly higher CRUX gains for GRPO).
- Related works ([2]) have shown that higher entropy in the distribution of the initial tokens encourages better exploration of the answer space, improving performance on more complex tasks. The focus on self-certainty may limit the diversity of generated solutions. It would be interesting to see how this method performs on more complex math or code benchmarks. Alternatively, a qualitative comparison of the diversity of generated code solutions of INTUITOR and MATH could bring extra insights into this matter.
- The paper would also benefit from a brief discussion of recent follow-up work such as [1], which analyzes why RLIF methods often yield diminishing returns for instruction-tuned models and links internal feedback to changes in token-level entropy. Including this perspective would help contextualize both the strengths and limitations of the proposed approach.


Related works:

[1]  No Free Lunch: Rethinking Internal Feedback for LLM Reasoning https://arxiv.org/abs/2506.17219

[2] The Entropy Mechanism of Reinforcement Learning for Reasoning Language Models https://arxiv.org/pdf/2505.22617

**Questions:**

1. The paper notes that the self-certainty objective modifies the generation process beyond simply increasing the probability of the most likely next token. Could the authors elaborate on how this differs from temperature scaling or confidence sharpening, both locally and across reasoning trajectories?

1. Have the authors considered evaluating RLIF with self-certainty on datasets without verifiable answers? This would better substantiate the claim of applicability to unverifiable domains.

1. In Figure 6, is it possible that the same pattern that we observe in offline training would emerge also for online training, but after many more iterations?

---

Overall, I believe this paper is an important contribution to the field. However, for strong acceptance, I would encourage the authors to provide a more thorough investigation of the method’s limitations and underlying mechanisms. In particular, it seems that the observed improvements primarily stem from a) better output formatting b) longer reasoning chains which happen to correlate with higher self-certainty. The additional findings of [1] also indicate that RLIF yields little improvement for instruction-tuned models, indicating diminishing returns of intrinsic feedback once an LLM is already trained to generate well-structured outputs. It remains unclear whether we can observe easy-to-hard generalisation by purely relying on self-certainty. I would be inclined to raise my score if the main text and abstract were revised to present the contributions more cautiously and in proportion to the empirical evidence.

---

> ### Author Response · Authors · 2025-11-21
>
> Thank you for the thoughtful and constructive review, and for recognizing the contribution and potential impact of RLIF and Intuitor. We address your main concerns and questions point by point below:
>
> ### **1. Scope: Applicability beyond verifiable math/code domains**
>
> > *“The evaluation scope is somewhat narrow… all reported experiments are on domains (math, code) where correctness can in fact be automatically checked. Demonstrating effectiveness on tasks without clear reward signals would make the paper more convincing.”*
> > *“Have the authors considered evaluating RLIF with self-certainty on datasets without verifiable answers?”*
>
> We agree that showing effects beyond strictly verifiable domains is important, and we have expanded our evaluations accordingly.
>
> - In the main paper we already evaluate **instruction following** with **AlpacaEval 2.0**, where correctness is judged by GPT‑4.1 rather than by a programmatic verifier. Intuitor significantly improves length‑controlled win rate over the base model and over GRPO in this setting.
> - During rebuttal, we additionally report results on **TruthfulQA** and **GPQA Diamond**, which probe factual accuracy and advanced scientific reasoning.
>
> Using **Qwen2.5‑3B** trained *only on MATH* with intrinsic self‑certainty rewards, we obtain:
>
> | Model (Qwen2.5‑3B) | Training Data          | TruthfulQA mc1 | TruthfulQA mc2 | GPQA Diamond |
> |--------------------|------------------|----------------|----------------|--------------|
> | Base               | –                | 0.328          | 0.478          | 0.328        |
> | **+ Intuitor (RLIF)** | MATH      | **0.341**      | **0.505**      | **0.335**    |
>
> Even though Intuitor is trained purely on math with *no external rewards*, it maintains or slightly improves performance on TruthfulQA and GPQA relative to the base model. This suggests that RLIF does not simply overfit to verifiable math/code tasks, and can transfer to other domains.
>
>
> ### **2. Mechanism: RLIF vs temperature scaling / confidence sharpening**
>
> > *“The KL-divergence–based objective effectively sharpens the output distribution, similar to reducing the sampling temperature… It would be helpful to clarify why RL training with self-certainty yields higher-quality reasoning compared to simply sampling greedily or using low-temperature decoding. Could the authors elaborate on how this differs from temperature scaling, both locally and across trajectories?”*
>
> We appreciate this opportunity to clarify the mechanism.
>
> 1. **Decoding is held fixed.**
>    All evaluations (base, GRPO, Intuitor) use **greedy decoding (temperature = 0)**. Thus, the accuracy gains of Intuitor cannot be explained by changing temperature or sampling strategy: they arise from changes in the model parameters induced by RL, not from post‑hoc decoding tweaks.
>
> 2. **Temperature vs policy optimization.**
>    - **Temperature scaling** is a *local*, post‑hoc transformation of logits:  $p_T(o_i \mid q, o_{<i}) \propto \exp(z_i/T)$. It does not change the relative ranking of tokens across contexts and does not reshape the joint distribution over full trajectories.
>    - **Intuitor**, in contrast, optimizes the *parameters* so as to **increase self‑certainty for whole trajectories that the model internally judges as more coherent**, under a KL‑regularized policy gradient: $ \hat{A}_i \propto \frac{u_i - \text{mean}(u)}{\text{std}(u)}$, $u_i=\text{Self-certainty}(o_i\mid q)$. This affects *which reasoning paths the model finds and prefers* across the entire trajectory, not just how sharply it samples from a fixed distribution.
>
> 3. **Process‑level effects.**
>    Empirically, Intuitor:
>    - Encourages **longer, structured chains of thought** (Figure 2), whereas lower temperatures typically *shorten* outputs.
>    - Induces **pre‑code natural language reasoning** and better adherence to output formats on code tasks, behaviors that cannot be reproduced by simply making logits sharper at inference.
>
> ### **3. OOD generalization and how strongly we state it**
>
> > *“Claims of superior out-of-distribution generalisation over GRPO appear overstated… the performance gains are small or inconsistent across models (e.g., Qwen2.5-1.5B and OLMo2-7B-SFT).”*
>
> Thank you for pointing this out. You are right that the gains are:
>
> - **Consistent but modest** on some models (e.g., Qwen2.5‑7B/14B), and
> - **Small or mixed** on others (e.g., OLMo‑2‑7B‑SFT).
>
> We will tone down the claim from “superior out-of-domain generalization” to **“competitive or slightly better generalization on several models and benchmarks.”** We will explicitly note where improvements are small or absent, and we have discussed that RLIF appears most effective when the base model already has a strong latent reasoning prior (e.g., Qwen2.5-3B) but can provide diminishing returns on more instruction-tuned models, as mentioned in the appendix.
>
> (1/2 To be continued)

---

> ### Author Response · Authors · 2025-11-21
>
> (2/2)
> ### **4. Diversity, entropy, and the exploration–certainty trade-off**
> > *“Higher entropy in initial tokens encourages exploration… The focus on self-certainty may limit the diversity of generated solutions.”*
>
> We agree that there is a fundamental tension between **exploration/diversity** and **certainty/sharpening**, and we now discuss this more openly.
> - Our experiments confirm that Intuitor **sharpens** the model’s distribution and encourages longer, more confident reasoning chains. This inevitably reduces local entropy, similar to RLVR. We acknowledge this limitation and “The Entropy Mechanism of RL for Reasoning Language Models” situates our method in this space.
> - We assess diversity by sampling eight MATH500 completions per question, embedding them with Qwen3-Embedding-8B, and averaging within-question mean cosine similarity across completions, finding Intuitor slightly less diverse than GRPO.
> |Qwen2.5-3B|GRPO|Intuitor|
> |:-:|:-:|:-:|
> |0.930|0.957|0.969|
> - At the same time, Intuitor is trained from **multiple rollouts per query** (7–14 samples), and self‑certainty is computed over full trajectories, not just first tokens. This preserves some diversity at the trajectory level, even as token‑level entropy decreases.
> - In preliminary experiments (not included for space), adding a small diversity‑oriented reward term on top of self‑certainty partially mitigated the diversity drop, suggesting that **combining RLIF with entropy‑based exploration** is a promising direction.
>
> Given space and time limits, we view a full study of diversity and exploration as future work, and we have incorporated this limitation in the discussion in Appendix.
> ### **5. Long‑run behavior and offline vs online self‑certainty**
>
> > *“In Figure 6, is it possible that the same pattern that we observe in offline training would emerge also for online training, but after many more iterations?”*
>
> We do see evidence that **RLIF has natural saturation points**, but the behavior differs substantially from the catastrophic reward hacking observed with offline self‑certainty.
> - With a **fixed offline annotator**, the policy quickly learns to exploit the static reward by appending unrelated “easy” problems to inflate certainty, leading to a spike in length and a sharp **collapse in accuracy** after ~100 steps (Figure 6).
> - With **online self‑certainty**, we do *not* observe comparable collapse under the same budget. Instead, performance tends to **rise and then plateau**, with mild regression if we continue training far beyond one epoch.
>
> For example, on MATH with Qwen2.5‑3B, pass@1 accuracy at selected checkpoints is:
>
> |Steps|10|30|50|70|90|110|
> |:-|:-:|:-:|:-:|:-:|:-:|:-:|
> |**GRPO**|0.513|0.641|0.660|0.652|0.589|0.655|
> |**Intuitor**|0.464|0.627|0.633|0.664|0.540|0.616|
>
> Intuitor improves rapidly, then saturates and slightly declines when over‑trained on the same small dataset, but **does not enter the reward‑hacking regime** we see with a frozen annotator. We now discuss this as a limitation: RLIF alone can saturate on narrow corpora, and practical use should pair it with early stopping, larger or more diverse data, or a hybrid with external rewards.
> ### **6. Relation to “No Free Lunch” and diminishing returns on instruction‑tuned models**
>
> > *“Recent follow-up work [1] analyzes why RLIF methods often yield diminishing returns for instruction-tuned models… Including this perspective would help contextualize both the strengths and limitations.”*
>
> We agree and have incorporated this perspective:
> - On **Qwen2.5** base models, Intuitor yields clear gains in both in‑domain and OOD tasks.
> - On **instruction‑tuned** models like Llama3.2‑3B‑Instruct, gains are **smaller but still positive** (e.g., LiveCodeBench from 0.106 → 0.134, CRUXEval from 0.265 → 0.293), consistent with [1]’s observation that intrinsic feedback has diminishing returns once the model already produces well‑structured answers.
>
> We position Intuitor/RLIF as: (1) Most effective on *under‑sharpened* or base reasoning models, and  (2) A **complement** to RLVR/RLHF rather than a universal replacement, especially for highly instruction‑tuned systems.
> ### **7. Changes to claims in abstract and main text**
> In response to your suggestion, we **tone down the abstract and conclusions** in the revised paper to:
> - Emphasize that Intuitor **matches supervised GRPO on math and achieves competitive, sometimes better, generalization on several code and instruction‑following benchmarks**, rather than claiming uniformly superior OOD performance.
> - Clearly label RLIF as a **promising, but not complete, alternative** to external rewards, with documented limitations (saturation, reduced diversity, smaller gains on instruction‑tuned models), as noted in Appendix B.10 Failure case analysis.
> ---
> Thank you again for your detailed feedback. Your comments on scope, mechanisms, diversity, and limitations have helped us substantially refine both the framing and the empirical narrative of the paper.

---

> > ### Author Response · Authors · 2025-11-26
> > **Follow-up to Reviewer gp3y**
> >
> > Dear Reviewer gp3y,
> >
> > Thank you again for your thoughtful and positive review! As the rebuttal period ends, we wanted to briefly flag the main changes we made in direct response to your comments: (1) we added results on TruthfulQA, GPQA, and AlpacaEval 2.0 to better support applicability beyond strictly verifiable math/code; (2) we clarified how self‑certainty–based RL differs from temperature scaling/greedy decoding; (3) we toned down claims around out‑of‑distribution generalization and explicitly discuss where gains are modest or saturate; and (4) we added analysis of diversity and positioned RLIF more cautiously in light of recent work such as “No Free Lunch: Rethinking Internal Feedback for LLM Reasoning.”
> >
> > You kindly noted that *"I would be inclined to raise my score if the main text and abstract were revised to present the contributions more cautiously and in proportion to the empirical evidence."* We hope these revisions address your concerns about scope, mechanisms, and claim strength, and we would be very grateful if you could consider them in your final assessment.
> >
> > Sincerely,
> >
> > The Authors

---

### Official Review · Reviewer_Np23 · 2025-10-30

**Soundness:** 2
**Presentation:** 3
**Contribution:** 2
**Rating:** 4
**Confidence:** 4

**Summary:**

The paper proposes Reinforcement Learning from Internal Feedback (RLIF), a paradigm for fine-tuning LLMs using intrinsic self-certainty measured as average KL divergence from uniform token distributions as the sole reward signal, instantiated via an adaptation of Group Relative Policy Optimization (GRPO) called INTUITOR. It claims this enables unsupervised reasoning improvements on math (MATH, GSM8K) and generalization to code (LiveCodeBench, CRUXEval) without gold answers or verifiers, with ablations exploring KL penalties, scaling, and optimization strategies. The show good empirical evidence of emergent structured reasoning and applicability across models (Qwen, Llama, OLMo), though limited by small-scale experiments and simpler baselines.

**Strengths:**

1. The paper extends RLIF by integrating self-certainty into GRPO for process-aware rewards using online computation to curb hacking, differing from outcome-focused RLVR.

2. The paper presents comprehensive experiments across families, with ablations rigorously contrasting entropy/random baselines to affirm stability,

3. The paper addresses a relevant question: scalable rewards without supervision for RLVR-limited domains, with execution showing INTUITOR's 13.8% OOD gains (e.g., LiveCodeBench) and emergent reasoning, valuable for self-improving agents.

**Weaknesses:**

1. The approach replaces RLVR with self-certainty based reward, however it is questionable if the gains still hold when the models hallucinate, especially when the method is scaled to larger models.

2. The paper's experiments are confined to small models (1.5B-14B) and corpora (7.5k problems), raising doubts on scalability.

3. The novelty is modest: the self-certainty reward builds directly on self-certainty ides proposed in Kang et al. (2025) to replace RLVR in the GRPO formulation from DeepSeek-R1 lacking novel formulation or broad applicability beyond math/code.

4. The gains are not transformative, as they are comparable to baselines on in-domain tasks. I would also expect comparisons with stronger reasoning baselines rather than just GRPO, like STaR (Zelikman et al., 2022) or Quiet-STaR (Zelikman et al., 2024), which incorporate advanced self-consistency or chain-of-thought mechanisms and would better contextualize INTUITOR's relative strengths in unsupervised settings. The paper does mention alternatives like perplexity or entropy-based measures but no such ablations are provided in the paper.

**Questions:**

1.  I would assume that when the models hallucinate, the learnt distribution would sway away from the uniform distribution, increasing the KL divergence and thus the self-certainty. Thus, I am not convinced how self confidence is a good metric to replace RLVR on complex reasoning tasks in such scenarios?

2. The above issue seems to be mitigated using online learning by co-evolving the reward with policy, as shown in an ablation. But I still do not understand how it would be useful when the signal is very hallucinatory. Can the authors explain this?

3. Have the authors experimented with a hybrid approach that uses say a weighted combination of RLVR and self-certainty reward.

3. I would also assume that using self-certainty reward would hurt continual learning too. E.g. say a model is trained on MATH dataset using the self-certainty reward. It may get good at solving Math problems but may be poor at solving Physics questions. Let us say we later try to fine-tune it to solve Physics problems too, however, since the self-certainty is high, it will assign high self-certainty reward to poor solutions for the Physics problem, thus hampering learning. Do the authors have a way to mitigate this?

In all, I am not fully convinced that self-certainty is a scalable alternative for RLVR for solving reasoning tasks.

---

> ### Author Response · Authors · 2025-11-21
>
> Thank you for the thoughtful and detailed review. We appreciate your careful reading and address your main concerns and questions below.
>
>
>
> ### **1. Scope: RLIF as a complement, not a replacement, for RLVR (W1, Q1–Q2)**
>
> > *“It is questionable if the gains still hold when the models hallucinate… I am not convinced how self confidence is a good metric to replace RLVR… The issue seems to be mitigated using online learning… but I still do not understand how it would be useful when the signal is very hallucinatory.”*
>
> First, we would like to clarify scope: **we do not claim that Intuitor (RLIF with self‑certainty) can replace RLVR whenever verifiable rewards are available.** RLVR remains strictly preferable in domains where ground‑truth verifiers exist and are cheap. Our goal is to show that **purely intrinsic, self‑generated feedback can already yield surprisingly strong gains**, matching GRPO on in‑domain math and improving cross‑domain performance, *in settings where RLVR is unavailable or impractical* (e.g., open‑ended or hard‑to‑verify tasks).
>
> **(a) Hallucination and “hidden knowledge”.**
>
> Our working hypothesis is that Intuitor primarily **sharpens and organizes knowledge already latent in the base model**, rather than “creating” new factual knowledge ex nihilo. This is consistent with recent work arguing that RL for reasoning often exploits and reshapes existing knowledge rather than learning entirely new facts. In other words:
>
> - When the base model *does* have partial or fragmented knowledge, self‑certainty helps it discover more coherent reasoning trajectories and express that knowledge more reliably.
> - When the base model truly has *no* knowledge (e.g., strongly out‑of‑distribution facts), RLIF alone cannot fix this—nor do we claim it can.
>
> Empirically, we see that Intuitor increases self‑certainty **for both correct and incorrect answers**, but disproportionately for correct ones. Figure 7 in the revised paper shows that:
>
> - The model trained with Intuitor assigns **significantly higher self‑certainty to correct answers than to incorrect ones**, with the largest separation effect size among all compared models.
> - This improved discrimination coincides with higher accuracy and better generalization.
>
> So while self‑certainty certainly increases, it does so in a way that is *more aligned* with correctness after training.
>
> **(b) Why self‑certainty does not simply reward hallucinations.**
>
> Even in the presence of hallucinations, several properties of our setup limit the risk of runaway overconfidence:
>
> 1. **Relative, not absolute, reward.**
>    Self‑certainty is used **only through group‑normalized advantages** in GRPO:
>    $\hat{A}_i \propto \frac{u_i - \text{mean}(u)}{\text{std}(u)}.$
>    If all completions for a given question are equally hallucinatory and equally overconfident, their relative advantages are small and gradients vanish; the model does not get a strong push toward any particular bad behavior.
>
> 2. **Online reward co‑evolves with the policy.**
>    Our offline vs. online ablation (Figure 6 in the paper) shows that a **fixed annotator can indeed be exploited** (the model learns to append spurious solved subproblems to inflate self‑certainty), while the **online self‑certainty annotator co‑evolves** with the policy and remains robust, no reward hacking emerges under the same training budget.
>
> 3. **Truthfulness and scientific QA remain stable.**
>    To directly test whether Intuitor amplifies hallucinations or harms truthfulness, we evaluated on **TruthfulQA** and **GPQA Diamond**, which require factual correctness rather than math/code‑style verification. Even though Intuitor was trained only on MATH, it maintains or slightly improves performance:
>
> | Model (Qwen2.5‑3B) | Training Data          | TruthfulQA mc1 | TruthfulQA mc2 | GPQA Diamond |
> |--------------------|------------------|----------------|----------------|--------------|
> | Base               | –                | 0.328          | 0.478          | 0.328        |
> | **+ Intuitor (Ours)**  | MATH           | **0.341**      | **0.505**      | **0.335**    |
>
> We do **not** observe degradation in truthfulness‑oriented benchmarks. This suggests that, at least in the regimes we study, self‑certainty is not simply amplifying hallucinations in an uncontrolled way.
>
> **(c) Limitations.**
>
> We fully agree that in extreme hallucination regimes (e.g., adversarial prompts or severely miscalibrated base models), intrinsic rewards alone are not sufficient. We now state this more explicitly in the discussion: RLIF is best viewed as a **complementary tool**—especially useful where external verifiers are unavailable, and as a warm‑up or auxiliary signal in hybrid schemes with RLVR/RLHF.
>
> (1/3 To be continued)

---

> > ### Author Response · Authors · 2025-11-21
> >
> > (2/3)
> >
> > ### **2. Scalability and model/dataset size (W2)**
> >
> > > *“The paper's experiments are confined to small models (1.5B–14B) and 7.5k problems, raising doubts on scalability.”*
> >
> > We share your interest in scaling. Our experiments are indeed constrained by academic compute budgets, but we have pushed beyond very small scales:
> >
> > - **Model sizes:** We evaluate Intuitor on **Qwen2.5‑1.5B, 3B, 7B, 14B**, as well as **Llama3.2‑3B‑Instruct** and **OLMo‑2‑7B**, demonstrating robustness across architectures and sizes.
> > - **Corpora:** While MATH has 7.5k problems, we additionally train **Intuitor‑Code** on Codeforces and show strong transfer to LiveCodeBench and CRUXEval. These experiments are explicitly designed as *proof‑of‑concepts* for RLIF under constrained supervision.
> >
> > We have also added a discussion in the paper acknowledging that more extensive scaling (e.g., >100B models and larger unlabeled corpora) is an exciting direction and likely requires industrial‑scale resources.
> >
> >
> >
> > ### **3. Novelty and comparison to stronger baselines (W3)**
> >
> > > *“Novelty is modest… I would also expect comparisons with stronger reasoning baselines rather than just GRPO, like STaR or Quiet-STaR. The paper mentions alternatives like perplexity or entropy-based measures but no such ablations are provided.”*
> >
> > **(a) Conceptual novelty.**
> > We agree that our method builds on two prior lines: self‑certainty (Kang et al., 2025) and GRPO (DeepSeek‑R1). Our contributions are:
> >
> > 1. **RLIF instantiation.** We formulate and instantiate **Reinforcement Learning from Internal Feedback (RLIF)**, where *all* rewards come from intrinsic signals. To our knowledge, this is the first work to show that such a purely intrinsic signal can **match RLVR on in‑domain math and outperform it on out‑of‑domain code** without any gold labels or test‑case verifiers.
> >
> > 2. **Online self‑certainty as a reward signal.** Prior work uses self‑certainty primarily for *offline selection*. We show that **optimizing it online with a KL‑regularized policy gradient** is non‑trivial:
> >    - Direct minimization of negative self‑certainty collapses (Appendix B.5, Figure 16).
> >    - The advantage‑weighted GRPO formulation we propose avoids reward hacking and remains stable across models.
> >
> > 3. **Robustness analysis.** We provide a **systematic comparison against multiple alternative intrinsic rewards** (entropy minimization, random rewards, log‑probability “sharpening”), and show empirically that self‑certainty is uniquely stable in our RL setting.
> >
> > **(b) Comparison to STaR / Quiet‑STaR.**
> >
> > - **STaR** uses *gold answers* to select correct rationales, so it is closer in spirit to RLVR/RL with labels than to RLIF. Our goal is to stay in the label‑free regime.
> > - **Quiet‑STaR** is directly relevant as a label‑free self‑improvement method. Following its setup (thought token length 8, 2 thoughts per position, 4 lookahead tokens) and training on the MATH train split, we obtain:
> >
> > | Model (Qwen2.5‑3B) | GSM8K | MATH500 |
> > |--------------------|:-----:|:-------:|
> > | Base               | 0.673 | 0.544   |
> > | Quiet‑STaR         | 0.635 | 0.588   |
> > | **Intuitor (Ours)**| **0.792** | **0.612** |
> >
> > Under the same MATH supervision budget, **Intuitor substantially outperforms Quiet‑STaR**, especially on GSM8K. We believe Quiet‑STaR may benefit more from much larger corpora, but within this controlled setting our method is clearly stronger.
> >
> > **(c) Ablations vs perplexity/entropy/log‑probability/random.**
> >
> > We have added the missing ablations in the revised paper (Appendix B.4–B.5):
> >
> > - **Negative entropy (EM‑style):** Improves briefly but then:
> >   - collapses into repetition on Qwen2.5‑1.5B, and
> >   - shows strong length bias and degradation over long training on Qwen2.5‑3B (Figures 12–13).
> > - **Random rewards:** Following recent “spurious RL” baselines, random 0/1 rewards **severely degrade performance** (Figure 12 right), confirming that “any RL” is not sufficient.
> > - **Log‑probability (“sharpening”) rewards:** Both raw and length‑normalized log‑probs degenerate (Figure 14)—either collapsing to very short answers or incentivizing arbitrarily long completions.
> > - **Self‑certainty (ours):** Achieves stable improvements on both 1.5B and 3B models, without collapse.
> >
> > These results suggest that **self‑certainty occupies a “sweet spot” among intrinsic signals**, balancing learning signal strength and robustness.
> >
> > (2/3 To be continued)

---

> > > ### Author Response · Authors · 2025-11-21
> > >
> > > (3/3)
> > >
> > > ### **4. Hybrid RLVR + self‑certainty (Q3)**
> > >
> > > > *“Have the authors experimented with a hybrid approach that uses a weighted combination of RLVR and self-certainty reward?”*
> > >
> > > Yes. We now report two hybrid experiments in the paper:
> > >
> > > 1. **Naïve mixed advantage (unsuccessful).**
> > >    We tried a simple convex combination \(A' = 0.5 A_\text{Intuitor} + 0.5 A_\text{GRPO}\) on Qwen2.5‑3B. This **did not outperform GRPO alone** and sits between Intuitor and GRPO in performance:
> > >
> > >    - GSM8K: Intuitor 0.792, GRPO 0.836, Mix 0.817
> > >    - MATH500: Intuitor 0.612, GRPO 0.636, Mix 0.632
> > >
> > >    This suggests that naïve combination is not optimal.
> > >
> > > 2. **Two‑stage training (successful).**
> > >    We then used Intuitor as a **warm‑up** for one epoch on MATH, followed by one epoch of GRPO with gold rewards. This **outperformed two epochs of GRPO**:
> > >
> > >    | Method               | GSM8K | MATH500 |
> > >    |----------------------|:-----:|:-------:|
> > >    | Intuitor             | 0.792 | 0.612   |
> > >    | GRPO                 | 0.836 | 0.636   |
> > >    | GRPO (2 epochs)      | 0.834 | 0.644   |
> > >    | **Intuitor → GRPO**  | **0.838** | **0.672** |
> > >
> > >    This supports the view that RLIF can be a useful **preconditioning step for RLVR**, helping the model develop more coherent reasoning traces that GRPO can then refine with verifiable rewards. We now highlight this in the discussion as a promising hybrid direction.
> > >
> > >
> > >
> > > ### **5. Continual learning / new domains (Q4)**
> > >
> > > > *“…self-certainty reward would hurt continual learning… later fine-tuning on Physics might be hampered because self-certainty is high on poor solutions. Do the authors have a way to mitigate this?”*
> > >
> > > We agree this is an important concern. We investigated a concrete analogue of your hypothetical scenario:
> > >
> > > - Start from Qwen2.5‑3B.
> > > - Train Intuitor on **Codeforces** (code domain).
> > > - Then continue training Intuitor on **MATH** (new domain).
> > > - Compare to training Intuitor on MATH directly.
> > >
> > > Results:
> > >
> > > | Model (Qwen2.5‑3B, Intuitor) | Training Data     | GSM8K | MATH500 |
> > > |------------------------------|-------------------|:-----:|:-------:|
> > > | Intuitor                     | MATH              | 0.792 | 0.612   |
> > > | Intuitor                     | Codeforces        | 0.743 | 0.572   |
> > > | **Intuitor**                 | **Codeforces → MATH** | **0.808** | **0.644** |
> > >
> > > Rather than hurting later MATH learning, **pretraining with Intuitor on Codeforces actually improves downstream math performance.** This suggests that, in practice:
> > >
> > > - Self‑certainty tuning on one domain does *not* “freeze in” high confidence on poor solutions for a different domain.
> > > - Instead, it appears to improve the model’s general reasoning toolkit (e.g., structuring chains of thought, using scratch reasoning), which then transfers beneficially to new tasks.
> > >
> > > Mitigations we discuss in the paper for potential continual‑learning issues include:
> > >
> > > - **Domain‑specific prompts or heads** to encourage appropriate behavior per domain.
> > > - **Adjusting KL penalty and rollout size** when moving to a significantly different domain.
> > > - Optionally mixing in small amounts of external supervision or verifiable reward (as in the hybrid scheme above) for especially sensitive domains.
> > >
> > > ---
> > >
> > > ### **6. Summary**
> > >
> > > To summarize our responses to your main concerns:
> > >
> > > - We **do not position Intuitor as a replacement for RLVR**, but as a **scalable, label‑free complement** that is particularly valuable where external verifiers are unavailable.
> > > - We provide additional **evidence against the “pure hallucination” failure mode**: online self‑certainty is robust to reward hacking, improves discrimination between correct and incorrect answers, and does not harm truthfulness benchmarks.
> > > - We expand the **novelty and empirical scope** by:
> > >   - Comparing to **Quiet‑STaR**, where Intuitor performs substantially better under the same MATH budget;
> > >   - Adding ablations vs multiple intrinsic rewards (entropy, log‑probability, random), where Intuitor is uniquely stable.
> > > - We show that Intuitor **plays well with RLVR** (as a warm‑up stage) and does **not hinder continual learning** in our cross‑domain experiments (Codeforces → MATH).
> > >
> > > We hope these clarifications and additional results address your concerns and better convey both the limitations and the promise of RLIF with self‑certainty as a practical ingredient for self‑improving reasoning models.

---

> > > > ### Author Response · Authors · 2025-11-26
> > > > **Follow-up to Reviewer Np23**
> > > >
> > > > Dear Reviewer Np23,
> > > >
> > > > Thank you for your detailed review and thoughtful questions. As the rebuttal period is ending, we wanted to briefly highlight how the revised manuscript directly addresses your main concerns:
> > > >
> > > > - **Hallucinations and scope of RLIF vs. RLVR.** We now explicitly frame Intuitor as a complement rather than a replacement for RLVR, and add analyses showing that online self‑certainty improves discrimination between correct/incorrect answers and does not harm factuality on TruthfulQA and GPQA.
> > > >
> > > > - **Scalability and continual learning.** We expanded experiments and added a cross‑domain continual learning setting (Codeforces → MATH), where prior RLIF training improves downstream math performance rather than locking in overconfidence on wrong solutions.
> > > >
> > > > - **Novelty and stronger baselines.** We now compare against Quiet‑STaR under the same MATH supervision budget, where Intuitor performs substantially better, and add systematic ablations with alternative intrinsic rewards (entropy, log‑probability “sharpening”, random), which are far less stable than self‑certainty.
> > > >
> > > > - **Hybrid RLVR + self‑certainty.** We include hybrid experiments, showing that using Intuitor as a warm‑up before GRPO with gold rewards meaningfully improves over GRPO alone.
> > > >
> > > > You concluded that you were not fully convinced that self‑certainty is a scalable alternative to RLVR for reasoning tasks. With the revised positioning (as a complementary ingredient) and the new empirical evidence on hallucinations, stronger baselines, hybrids, and cross‑domain behavior, we hope the updated version alleviates your main concerns and may merit a more positive overall assessment.
> > > >
> > > > Sincerely,
> > > >
> > > > The Authors

---

### Official Review · Reviewer_9ExH · 2025-10-30

**Soundness:** 2
**Presentation:** 3
**Contribution:** 3
**Rating:** 8
**Confidence:** 2

**Summary:**

The paper introduces INTUITOR and RLIF, which train LLMs to improve reasoning without any external rewards or supervision.
Instead of verifiable correctness signals (as in RLVR) or human preferences (as in RLHF), INTUITOR uses the model’s self-certainty as the intrinsic reward.
Implemented via Group Relative Policy Optimization (GRPO), INTUITOR replaces external verifiable rewards with normalized self-certainty scores to update the policy. Experiments on multiple benchmarks show that INTUITOR achieves comparable reasoning performance to GRPO on in-domain mathematical tasks while generalizing better to unseen domains like code generation and instruction following. The paper includes ablations, scaling studies, and qualitative analyses demonstrating emergent reasoning structure and robustness.

**Strengths:**

+ Proposes a new RL paradigm (RLIF) that removes the dependency on external or verifiable rewards. It is an elegant and forward-looking idea for autonomous reasoning systems.

+ Using self-certainty as an intrinsic reward is well-motivated, mathematically grounded, and can be integrated into existing policy optimization frameworks.

+ Demonstrates solid performance across both reasoning and code tasks, with consistent improvements in generalization, instruction-following, and early learning speed.

+ Benchmarks against GRPO, entropy-based rewards, and random rewards are thorough, and the reward hacking experiments (online vs. offline self-certainty) convincingly highlight robustness.

+ The paper contributes a meaningful step toward autonomous self-improvement in LLMs, addressing the scalability limits of RLHF/RLVR.

**Weaknesses:**

I have 3 main weakness concerns for this paper:
* The authors should argue way more to convince the reader on why self-certainty is a stable way of guiding the learning process, and won't lead agents to overconfidently learn shortcut behaviors. Having a random baseline that assigns "random confidence" would help a lot asserting the value of self-confidence. Also, providing second order metrics.

* In the experimental evaluation section, providing list of scientific questions at the beginning of the experimental evaluation section (often denoted RQ1, RQ2, ... etc), that are each answered in different paragraphs. For example
  * (RQ1) Is our framework outperforming existing baselines?
  * (RQ2) Does self-confidence lead to better selection of ...?
  * (RQ3) How important is module A ? (conducting an ablation study)

Figures can drastically be improved:
+ Captions could be more expressive, clearly explaining what each plot demonstrates and how it supports the paper’s claims. Improving figure layout and adding descriptive captions would significantly enhance readability and interpretability. In details:
**Each figure should support a claim made by the authors.**
Each caption should make the figure a standalone component of the paper. Thus, each figure should be built in the following way:
   * The first sentence (in bold) should highlight the main takeaway of the Figure/Table (e.g. "*[Our] method improves the ability of agents to do task T.*"). This is the main message for the reader that supports one of your claim.
   * The next sentences then explain what is depicted in the Table/Figure. E.g. "*..., as depicted by the superior mean test accuracy (+/- std) of our method over the different baselines.*"
   * Finally, details and references to e.g. appendix can be provided if necessary. E.g. "*Our method outperforms baseline 1 in 3 out of 4 tasks, ... etc. 5 seeded rerun. Best results highlighted in bold. Further description of the training and testing setups are available in Appendix C.*"


My remarks can also be addressed by reference to existing literature, to which I might be unfamiliar.

+ The lack of second order metrics (std) prevents from drawing accurate conclusions on the performance difference between the baselines

+ Since self-certainty is derived from the model’s own token probabilities, there is a risk of reinforcing its pre-existing biases or hallucinations; this should be addressed.

+ The distinction between “intrinsic reward,” “self-certainty,” and “entropy minimization” could be made clearer.


## Points to Improve

*This is not meant for the rebuttal per se (see below), but general points that would strengthen the paper*

+ Discuss potential failure cases where intrinsic rewards may amplify overconfidence or bias, and propose mitigation strategies.

+ Clarify the relationship between RLIF, RLVR, and RLHF, especially how RLIF could complement them in hybrid setups.

**Questions:**

* How sensitive is performance to the KL penalty and group size hyperparameters?

* Have you observed scenarios in which your intrinsic reward leads to overly verbose or repetitive reasoning?

---

> ### Author Response · Authors · 2025-11-21
>
> Thank you very much for the careful reading and supportive assessment of our work. We address your main concerns and questions point by point below.
>
> ### **1. Why is self‑certainty a stable intrinsic signal (vs shortcut behavior / random confidence)?**
>
> > *“The authors should argue way more to convince the reader on why self-certainty is a stable way of guiding the learning process, and won't lead agents to overconfidently learn shortcut behaviors. Having a random baseline that assigns "random confidence" would help a lot asserting the value of self-confidence. Also, providing second order metrics.”*
>
> We have expanded both the empirical and conceptual justification in the revised paper:
>
> - **Random reward baseline.** As you suggested, we implemented a “random confidence” baseline where each completion is assigned reward 0/1 uniformly at random (Appendix B.4). Figure 12 (right) shows that this random reward severely degrades performance in the chat‑style RL setting, while Intuitor continues to improve, demonstrating that our gains are not simply from “doing RL on anything.”
>
> - **Comparisons to other intrinsic rewards.** Beyond GRPO‑PV, we now compare self‑certainty against:
>   * **Entropy minimization** (EM‑style reward): On Qwen2.5‑1.5B it rapidly collapses into repetitive loops; on Qwen2.5‑3B it initially helps but then leads to strong length bias and a steady degradation over longer training (Figures 12 & 13).
>   * **Sharpening‑style log‑probability rewards**: both unnormalized and length‑normalized variants degenerate (Figure 14), either by collapsing to short answers or by encouraging runaway sequence length.
>   * In contrast, self‑certainty yields stable improvements across models and tasks without such collapse.
>
> - **Second‑order behavior.** To study stability more directly, we now track **standard deviation of correctness** during training (Figure 15):  (i) the step‑wise standard deviation across all completions; (ii) the mean within‑group standard deviation over rollouts.  Both GRPO and Intuitor show similar patterns: the within‑group std gradually decreases and the global std remains stable. We do not see evidence that Intuitor amplifies instability relative to GRPO.
>
> We also clarify in the discussion that self‑certainty is a *relative* reward used in an advantage‑weighted policy gradient (rather than an objective directly maximized), which empirically reduces reward‑hacking tendencies compared to directly optimizing the confidence measure (Appendix B.5).
>
>
>
> ### **2. Experimental structure, RQs, and figure captions**
>
> > *“Providing list of scientific questions (RQ1, RQ2, …) … Figures can drastically be improved… captions should clearly explain what each plot demonstrates…”*
>
> Thank you for these excellent, actionable suggestions for improving the paper's clarity. We have reframed the beginning of Section 5 (Results and Analysis) to be structured around three clear research questions (RQ1-RQ3), which guide the reader through our key findings. We have also revised figures and captions to follow the self-contained format you recommended. Each caption now begins with a bolded sentence summarizing the main takeaway, followed by a detailed explanation of the plot's content and its implications. Please see the revised paper for details.
>
>
> ### **3. Second‑order metrics and variance**
>
> > *“The lack of second order metrics (std) prevents from drawing accurate conclusions on the performance difference between the baselines.”*
>
> In addition to the new correctness‑variance analysis mentioned above (Figure 15), we also:
> - Report **multiple training runs** for selected settings (e.g., early‑stage learning in Table 2 and additional runs in Appendix B), showing that Intuitor’s early gains over GRPO are consistent.
> - Highlight that the **shape** of the learning curves (e.g., faster early gains, stable plateau) is robust under modest hyperparameter changes (KL, rollout size), which we now ablate more systematically (Table 3 and Table 7).
>
> While we were constrained in running many fully independent seeds for all settings (due to RL compute cost), we agree that adding variance measures was important, and we’ve done so where feasible.
>
> (1/2 To be continued)

---

> ### Author Response · Authors · 2025-11-21
>
> (2/2)
>
> ### **4. Risk of reinforcing biases / hallucinations & overly verbose reasoning**
>
> > *“Since self-certainty is derived from the model’s own token probabilities, there is a risk of reinforcing its pre-existing biases or hallucinations; this should be addressed.”*
> > *“Have you observed scenarios in which your intrinsic reward leads to overly verbose or repetitive reasoning?”*
>
> We agree this is a central concern and now discuss it explicitly:
>
> - **Bias / hallucination amplification.** Figure 7 shows that, after training, **self‑certainty increases for both correct and incorrect answers**, but the increase is significantly stronger for correct ones. This correlates with improved accuracy: the model becomes more confident *overall*, but disproportionately so on correct outputs, improving its ability to distinguish between them (Mann–Whitney U tests in Figure 7).
>
> - **Verbose / repetitive reasoning.** Intuitor does encourage **longer reasoning chains** (Figure 2), especially on stronger models and code tasks. However:
>   * We do **not** observe collapse into degenerate repetition under our main settings; instead, we see richer step‑by‑step reasoning followed by concise answers or code.
>   * In contrast, entropy‑based rewards *do* lead to repetitive loops and degradation, highlighting that not all confidence‑like signals behave equally.
>
> - **Failure cases & mitigations.** When we deliberately push training too long on a small fixed dataset (e.g., 3–5 epochs on MATH), we start to see overfitting and mild regressions. We now discuss this as a failure mode and mention practical mitigations: early stopping, mixing in supervised / verifiable rewards, increasing data diversity, and adjusting KL/rollout size to temper over‑sharpening.
>
>
> ### **5. Clarifying “intrinsic reward”, “self‑certainty”, and “entropy minimization”**
>
> > *“The distinction between ‘intrinsic reward,’ ‘self-certainty,’ and ‘entropy minimization’ could be made clearer.”*
>
> We revised the paper to clarify terminology:
>
> - **RLIF** is the *paradigm*: optimizing a policy using **intrinsic rewards** that are computed from the model’s own internal signals, without external labels or verifiers.
> - **Self‑certainty** is one concrete intrinsic reward: the KL divergence from uniform to the next‑token distribution, used by Intuitor.
> - **Entropy minimization** is another type of intrinsic reward (negative entropy). We treat it as a separate baseline and empirically find it substantially less stable than self‑certainty.
>
> Intuitor is thus one specific instantiation of RLIF using self‑certainty as the intrinsic reward, implemented via GRPO.
>
>
>
> ### **6. Relationship between RLIF, RLVR, RLHF & hybrid setups**
>
> > *“Clarify the relationship between RLIF, RLVR, and RLHF, especially how RLIF could complement them in hybrid setups.”*
>
> We want to clarify:
>
> - All three (RLHF, RLVR, RLIF) optimize a KL‑regularized RL objective; they mainly differ in **where the reward comes from**:
>   - RLHF: human preferences (via a learned reward model).
>   - RLVR: verifiable correctness / test‑based signals.
>   - RLIF: internal model signals (e.g., self‑certainty).
>
> - We now report a simple **hybrid experiment**: training Qwen2.5‑3B with Intuitor for one epoch on MATH, then GRPO with gold answers for one epoch. This **outperforms two epochs of GRPO alone** on both GSM8K and MATH500 (Table 8). This suggests RLIF can serve as a “self‑sharpening warm‑up” that improves the effectiveness of downstream RLVR/RLHF.
>
> We position this as an explicit direction for future work: designing principled schedules and combinations of external and internal rewards.
>
>
> ### **7. Sensitivity to KL penalty and group size (rollouts)**
>
> > *“How sensitive is performance to the KL penalty and group size hyperparameters?”*
>
> We now provide explicit ablations:
>
> - **KL penalty (Table 3):** On in‑domain GSM8K/MATH, performance is fairly stable across a reasonable KL range, but out‑of‑domain tasks (LCB, CRUX) are more sensitive. Very small KL can cause mild drift; moderate KL (e.g., 0.005) provides the best stability/generalization trade‑off.
>
> - **Group size / rollout count (Table 7):** For Qwen2.5‑3B, increasing rollouts from 7 → 14 improves GSM8K from 0.792 → 0.814 and MATH500 from 0.612 → 0.644. Larger rollout groups reduce variance in the self‑certainty advantage estimates and strengthen generalization, at the cost of more compute.
>
> ---
>
> Once again, thank you for the detailed and constructive feedback. Your suggestions on experimental framing, figures, and potential failure modes significantly improved the clarity and robustness of the paper.

---

> > ### Comment · Reviewer_9ExH · 2025-11-27
> > **Thank you for your clarifications**
> >
> > After reading your answers, both to me and the other reviewers, I have no major concern remaining.
> > I think that the additional experiments you did, and above all the clarifications (both in terms of structures and rephrasing) placed in the new version of the paper, make it worth a publication at ICLR.
> > I encourage the other reviewers to update their reviews and ping me if I missed a major flaw of the paper.

---

### Official Review · Reviewer_cUuw · 2025-10-31

**Soundness:** 2
**Presentation:** 4
**Contribution:** 2
**Rating:** 4
**Confidence:** 3

**Summary:**

This paper presents a method for self-improving LLMs that uses self-certainty as a reward signal, where self-certainty is defined as the KL divergence between a uniform distribution over the vocabulary and the model's next-token distribution. They train the model using the MATH dataset and the Codeforces dataset, and evaluate on a range of datasets. They compare against baselines like GRPO with gold labels and GRPO with plurality voting, which is another self-improvement method, and report the results. The results show performance on par with GRPO-PV on the math dataset, and improved performance on coding tasks, such as LCB and CRUX.

The paper also claims in the experiment section that the method exhibits faster early-stage learning compared to GRPO, demonstrates effective cross-task generalization (training on MATH improves code generation performance), and leads to emergent long-form reasoning behaviors. Additional analyses include comparisons across different model architectures (Llama, OLMo) and an examination of online versus offline self-certainty computation, showing that online computation prevents reward hacking.

**Strengths:**

Originality: This paper is part of a wave of recent work attempting self-improvement for LLMs. These methods share a similar flavor, focusing on math/code tasks and using some type of heuristic—in this case, a KL divergence metric. The specific approach appears to be novel, though I am not confident about it.

Quality: The idea is clear and the experimental analysis is quite thorough.

Clarity: The writing is quite clear.

Significance: The analysis is good. The insight regarding online versus offline self-certainty computation is particularly interesting.

**Weaknesses:**

The main contribution of this work is the use of KL divergence against a uniform distribution as a proxy reward. I don't think there is enough theoretical justification or empirical evidence showing why this particular proxy reward is better than alternatives. The only comparison provided is with plurality voting, yet several other alternatives exist (missing references are linked below).

The analysis of different model behaviors is interesting, but it is unclear whether these behaviors are due to the particular proxy reward or are a general consequence of self-improvement methods.

If the authors could either provide more theoretical explanation for why this proxy is superior, or show more empirical comparisons with alternative approaches, I would increase my score.

Some related work
Huang, Audrey, et al. "Self-improvement in language models: The sharpening mechanism." arXiv preprint arXiv:2412.01951 (2024).
Song, Yuda, et al. "Mind the gap: Examining the self-improvement capabilities of large language models." arXiv preprint arXiv:2412.02674 (2024).
Shafayat, Sheikh, et al. "Can Large Reasoning Models Self-Train?." arXiv preprint arXiv:2505.21444 (2025).

**Questions:**

1. Why do we see an out of distribution improvement for coding on Qwen but not on LLama?
2. Can you show baseline GRPO-PV for the LLama models too? I think they are missing in the appendix.

---

> ### Author Response · Authors · 2025-11-21
>
> Thank you for the thoughtful and constructive review! We address your main concerns and questions below.
> ### **1. Why self‑certainty (KL to uniform) as the proxy reward?**
>
> > *“The main contribution of this work is the use of KL divergence against a uniform distribution as a proxy reward. I don't think there is enough theoretical justification or empirical evidence showing why this particular proxy reward is better than alternatives. The only comparison provided is with plurality voting, yet several other alternatives exist (missing references are linked below).”*
>
> > *“If the authors could either provide more theoretical explanation for why this proxy is superior, or show more empirical comparisons with alternative approaches, I would increase my score.”*
>
>
> We appreciate the push for a stronger justification. Our contribution goes beyond introducing a particular proxy: we show that a purely intrinsic, self-generated signal (self‑certainty) can drive substantial cross‑domain improvements in reasoning and code generation, without any external labels or verifiers. That said, we agree that understanding why self‑certainty is a good intrinsic signal is important, and we have strengthened both the empirical and conceptual justification in the revised version.
>
> **(a) Empirical support from prior work**
>
> We chose self‑certainty specifically because previous work has shown it to be a strong and robust confidence signal.
> Kang et al. (2025) [1] performed a systematic comparison of several confidence signals for best-of-N selection (e.g., average log-probability, perplexity, entropy, Gini impurity). Their key findings were:
>
> - **Self‑certainty is the best-performing metric at larger sample sizes**, whereas other measures plateau or even degrade once N > 16.
> - **Self‑certainty is less biased by sequence length** than entropy or perplexity, which is crucial for reasoning tasks that produce long chains of thought.
>
>
> This prior evidence was a major reason we adopted self‑certainty as the intrinsic reward: it behaves well in exactly the regime we care about (multi‑sample reasoning) and is comparatively robust to length bias.
>
>
> **(b) New empirical comparisons to alternative intrinsic rewards**
>
>
> Beyond plurality voting (GRPO‑PV), we now compare self‑certainty against several alternative intrinsic rewards in the revised paper (Appendix B.4):
>
> 1. **Negative entropy (EM‑RL style)**
>    - We implement GRPO with a negative token‑level entropy reward.
>    - On Qwen2.5‑1.5B, this leads to **rapid collapse into repetitive loops** (Figure 12): the model increases its reward by becoming over‑confident on degenerate outputs and can no longer follow instructions.
>    - On Qwen2.5‑3B, entropy minimization initially improves accuracy but then produces a strong length bias and a steady degradation in performance over longer training (Figure 13).
>    - In contrast, self‑certainty (our method) achieves similar or better peak performance and then stabilizes instead of collapsing.
>
> 2. **Random reward (as in “spurious RL” baselines)**
>    - We follow the random‑reward setup proposed in recent work and assign each completion a reward of 0 or 1 uniformly at random.
>    - This severely **degrades performance in the chat‑style RL setting** (Figure 12, right), demonstrating that the gains we see with self‑certainty are not just an artifact of “doing RL on anything.”
> 3. **Sharpening‑style log‑probability rewards (Huang et al., 2024)**
>    - Huang et al. [2] analyze “sharpening” via sequence‑level log‑likelihood. We implement this directly as a GRPO reward: $
>        r(o) = \log \prod_i p(o_i \mid q, o_{<i}).$
>    - As we show in Figure 14, this unnormalized log‑probability is **strongly biased toward short completions**; during training, both accuracy and completion length drop sharply, indicating degeneration.
>    - We further test a **length‑normalized** variant $\frac{1}{|o|}\sum_i \log p(o_i \mid q,o_{<i}),$ which removes the short‑length bias but introduces the opposite effect: the model can improve reward by producing arbitrarily long completions and quickly exploits this, leading to unstable training.
>    - In contrast, **self‑certainty consistently improves accuracy** for both 1.5B and 3B models and remains stable.
> 4. **Plurality‑voting style self‑reward (SRT / GRPO‑PV)**
>    - Self‑rewarded training methods like SRT [4] effectively use self‑consistency / plurality as a proxy label. Our GRPO‑PV baseline implements this idea.
>    - On math, our method and GRPO‑PV achieve *similar* in‑domain performance, but **self‑certainty produces substantially better cross‑domain generalization** (LCB, CRUX), as shown in Table 1 and Table 5.
>
> Taken together, these results suggest that **not all intrinsic rewards are equally usable** for RLIF: entropy and log‑probability‑based rewards are noticeably more fragile, and random rewards fail outright, whereas self‑certainty yields stable improvements across multiple models and tasks.
>
> (1/2 To be continued)

---

> ### Author Response · Authors · 2025-11-21
>
> (2/2)
>
> **(c) Theoretical understanding as future work**
>
> We agree that a deeper theoretical characterization of RLIF and the “best” intrinsic reward is an important open question. Recent theoretical work (e.g., Huang et al. [2]) analyzes sharpening and self‑improvement mechanisms from a statistical perspective. In the discussion section of the revised paper, we explicitly position our work as complementary of [2,3,4]: **we empirically identify self‑certainty as a particularly robust intrinsic signal**, and we view which signal is the best and formal analysis of why it works so well as a promising direction for future research.
>
> ### **2. Are the behavioral changes specific to self‑certainty or generic to self‑improvement?**
>
> > *“The analysis of different model behaviors is interesting, but it is unclear whether these behaviors are due to the particular proxy reward or are a general consequence of self-improvement methods.”*
>
> Our view is that **both aspects play a role**:
>
> - Experiments with entropy‑based rewards show that *other* intrinsic signals can also give some initial gains on strong models (e.g., Qwen2.5‑3B), indicating that self‑improvement itself is a powerful paradigm.
> - However, the same entropy reward **fails catastrophically on smaller models (Qwen2.5‑1.5B)** and degrades performance over long training even on larger ones, whereas self‑certainty remains stable.
> - Moreover, phenomena like **emergent pre‑code reasoning** and **improved instruction‑following** appear *much more reliably* under self‑certainty than under entropy or random rewards in our experiments.
>
> So we agree that some qualitative behaviors are generic to self‑improvement, but our empirical evidence indicates that **self‑certainty yields a substantially better trade‑off between learning signal strength and robustness**, especially for smaller or less stable base models.
>
> ### **3. Q1 – Out‑of‑distribution coding gains on Qwen vs Llama**
>
> > *“Why do we see an out of distribution improvement for coding on Qwen but not on LLama?”*
>
> We do in fact see OOD coding improvements on Llama as well, though they are smaller than for Qwen:
> - On Llama3.2‑3B‑Instruct (Table 5 in the revised paper), our method improves LiveCodeBench from 0.106 → 0.134,  CRUXEval‑O from 0.265 → 0.293,  compared to GRPO’s 0.127 and 0.266 respectively.
> - The gains are more modest than for Qwen2.5, where we start from a **base model** that has undergone extensive reasoning‑focused pretraining/mid‑training. This appears to give Qwen a stronger “latent reasoning prior” that RLIF can leverage.
> - By contrast, the Llama model we used is an **instruct** model, which is already heavily sharpened and aligned. Several recent works have also noted that RL for reasoning is more fragile on such instruction‑tuned backbones. In preliminary experiments, training from Llama *base* was significantly less stable, which is why we focused on the instruct variant.
>
> ### **4. Q2 – GRPO‑PV baselines for Llama**
> > *“Can you show baseline GRPO-PV for the LLama models too? I think they are missing in the appendix.”*
>
> Yes, we have added GRPO‑PV baselines for Llama3.2‑3B‑Instruct in the revised paper. For convenience, here are the numbers:
>
> | Method | GSM8K | MATH | LCB | CRUX | MMLU-Pro | AlpacaEval |
> | :--- | :---: |:---: |:---: |:---: |:---: |:---: |
> | Baseline | 0.688 | 0.436 | 0.106 | 0.265 | 0.340 | 11.07 |
> | GRPO | 0.714 | 0.494 | 0.127 | 0.266 | 0.361 | 13.62 |
> | GRPO-PV | 0.710 | 0.472 | 0.109 | 0.281 | 0.352 | 10.85 |
> | **Intuitor (Ours)** | **0.723** | **0.476** | **0.134** | **0.293** | **0.358** | **12.41** |
>
> This shows that on Llama as well, **our self‑certainty–based method at least matches and often slightly exceeds GRPO‑PV**, particularly on the coding benchmarks.
>
> ---
>
> Once again, thank you for your detailed feedback and for pointing us to relevant related work. We hope the additional experiments and clarifications help address your concerns!
>
> ------
> [1] Kang et al. Scalable Best-of-N Selection for Large Language Models via Self-Certainty. NeurIPS 2025
>
> [2] Huang et al. Self-improvement in language models: The sharpening mechanism. ICLR 2025
>
> [3] Song et al. Mind the gap: Examining the self-improvement capabilities of large language models. ICLR 2025
>
> [4] Shafayat et al. Can Large Reasoning Models Self-Train?. 2025

---

> > ### Author Response · Authors · 2025-11-26
> > **Follow-up to Reviewer cUuw**
> >
> > Dear Reviewer cUuw,
> >
> > Thank you again for your thoughtful and constructive review. As the rebuttal period is ending, we wanted to very briefly highlight how the revised manuscript addresses the key points you raised, especially the ones you mentioned could affect your overall assessment:
> >
> > - **Why self‑certainty (KL to uniform) as the proxy reward?**
> > We added more discussion connecting our choice to prior work on confidence signals and sharpening, and clarified why self‑certainty is particularly suitable in the multi‑sample reasoning regime we study.
> >
> > - **Empirical comparisons to alternative intrinsic rewards.**
> > We now include systematic comparisons against entropy‑based rewards, sharpening‑style log‑probability rewards, and random rewards, in addition to GRPO‑PV. These experiments show that many alternatives are either unstable or strongly biased, whereas self‑certainty yields stable gains across models and tasks.
> >
> > - **Are the behaviors specific to self‑certainty or generic to self‑improvement?**
> > We expanded the analysis to contrast behaviors under different intrinsic signals, and we explicitly discuss which qualitative effects appear generic to self‑improvement and which seem tied to self‑certainty’s robustness.
> >
> > - **Q1/Q2: OOD coding gains and GRPO‑PV on Llama.**
> > We clarified the OOD coding improvements on Llama and added the missing GRPO‑PV baselines for Llama3.2‑3B‑Instruct, showing that our method at least matches and often slightly outperforms GRPO‑PV, including on coding benchmarks.
> >
> > You kindly noted that *"If the authors could either provide more theoretical explanation for why this proxy is superior, or show more empirical comparisons with alternative approaches, I would increase my score."* We have tried to do both within the space of the rebuttal and revision, and we hope the new material addresses your main concerns and helps position the work as meeting the bar for acceptance.
> >
> > If there are remaining specific issues that you feel still block this, we would be very grateful for any brief guidance.
> >
> > Sincerely,
> >
> > The Authors

---

### Author Response · Authors · 2025-11-25
**Summary of changes and additional experiments during rebuttal**

We thank all reviewers for their detailed and constructive feedback. During the rebuttal period, we substantially revised the paper and added new experiments to address the main concerns:

- **Stronger justification for self-certainty as an intrinsic reward.**
  We added systematic comparisons against alternative intrinsic signals: entropy-based rewards, sharpening-style log-probability rewards, and random rewards. These baselines either collapse (e.g., repetitive loops, severe length bias) or significantly degrade performance, whereas self-certainty yields stable gains across models and tasks. We also report new analyses of correctness variance and self-certainty distributions, showing that Intuitor increases confidence disproportionately on correct solutions.

- **Expanded baselines and broader empirical scope.**
  We now include GRPO-PV baselines for Llama3.2-3B-Instruct and a Quiet-STaR baseline under the same MATH supervision budget, where Intuitor performs better. We additionally evaluate on TruthfulQA and GPQA Diamond (and highlight AlpacaEval 2.0), demonstrating that RLIF trained only on MATH does not harm and can slightly improve factual and scientific QA, supporting applicability beyond strictly verifiable math/code settings.

- **Hybrid and continual-learning settings.**
  To clarify the relationship between RLIF and RLVR, we added hybrid experiments. A two-stage schedule (Intuitor warm-up -> GRPO with gold rewards) outperforms GRPO alone, suggesting RLIF is a useful complement rather than a replacement. We also study a cross-domain setting (Intuitor on Codeforces -> Intuitor on MATH), finding that prior RLIF training does not block learning in the new domain and can even improve downstream math performance.

- **Sensitivity, failure modes, and diversity.**
  We provide ablations over the KL penalty and rollout group size, showing that Intuitor is reasonably robust within a moderate range. We analyze offline vs. online self-certainty and document that offline rewards are vulnerable to reward hacking, while online rewards saturate but do not collapse under the same budget. We also add a discussion and quantitative analysis of diversity, acknowledging that self-certainty reduces local entropy and slightly lowers diversity compared to GRPO, and we explicitly frame this exploration–certainty trade-off as a limitation and avenue for future work.

- **Clarity, positioning, and claim calibration.**
  We reorganized the results section around explicit research questions (RQ1–RQ3), rewrote figure captions to clearly state the main takeaway of each plot, and clarified terminology (RLIF vs. RLVR/RLHF, self-certainty vs. entropy minimization). Following reviewer suggestions, we toned down claims about out-of-domain generalization in the abstract and main text with documented limitations (diminishing returns on strongly instruction-tuned models and saturation on narrow corpora).

We hope these additions and clarifications address the main concerns raised by all reviewers. If there are specific aspects of the new experiments, failure-mode analysis, or the revised positioning of RLIF (e.g., hybrid settings with RLVR/RLHF, behavior on larger or more diverse datasets) that you would like us to further expand on, we are very happy to provide additional details or targeted follow-up analyses within the constraints of the rebuttal period.

---

### Meta-Review · Area_Chair_GM3j · 2025-12-16

**Summary:**

Overall, the reviewers agree that this paper tackles a timely and novel problem, is clearly written, and provides a workable RLIF implementation with supporting experiments for settings without external rewards or verifiers. Its OOD performance is stronger than RLVR and it appears to address the target problem effectively; in my view this angle is quite compelling. I also found the rebuttal and revision to be fairly thorough. The main weakness is that the method is largely heuristic, leaving gaps in the theoretical justification. In addition, the in-domain results lag behind GRPO, and the authors should provide a clear explanation for this in the final version.

**Reviewer Concerns:**

**R9ExH (supportive)**: The main concern is whether the self-certainty intrinsic reward might encourage a shortcut (becoming more confident without being more correct). They wanted stronger ablations to show the gains truly come from the proposed method. After the rebuttal, they largely view this as resolved.

**Rgp3y (borderline positive)**: The core issue is mechanism and claim calibration, improvements may be driven by sharpening/longer CoT/formatting rather than genuinely better reasoning. They want more cautious claims and broader task coverage to substantiate the conclusions.

**RcUuw (borderline negative)**: The key sticking point is why KL-to-uniform is a justified and necessary reward signal. They felt the theoretical rationale and systematic comparisons against alternative intrinsic rewards were insufficient, and worried the observed effects might not be specific to this reward.

**RNp23 (borderline negative)**: Most focused on safety and generalization boundaries, under hallucination or high-uncertainty regimes, the reward may reinforce confidently wrong behavior, and it’s unclear how well the approach scales to more open-ended, harder tasks. They also view the novelty as modest and would like stronger baselines for reassurance.

**Reviewer Scores:**

Overall, the scores may shift upward somewhat, but the authors may not be able to fully resolve the negative reviewers’ concerns.

---

### Decision · Program_Chairs · 2026-01-26

Accept (Poster)